


# Properties and emission factors of CCN from biomass cookstoves - observations of a strong dependency on potassium content in the fuel

Thomas Bjerring Kristensen[1], John Falk[1], Robert Lindgren[2], Christina Andersen[3], Vilhelm B. Malmborg[3], Axel C. Eriksson[3], Kimmo Korhonen[4], Ricardo Luis Carvalho[2,5], Christoffer Boman[2], Joakim Pagels[3], and Birgitta Svenningsson[1]

[1]Department of Physics, Lund University, SE-22100, Lund, Sweden
[2]Thermochemical Energy Conversion Laboratory, Department of Applied Physics and Electronics, Umeå University, SE-90187, Umeå, Sweden
[3]Ergonomics and Aerosol Technology, Lund University, SE-22100, Lund, Sweden
[4]Department of Applied Physics, University of Eastern Finland, FI-70211, Kuopio, Finland
[5]Centre of Environment and Marine Studies, University of Aveiro, PT-3810-193 Aveiro, Portugal

**Correspondence:** Thomas B. Kristensen (thomas.bjerring_kristensen@nuclear.lu.se)

**Abstract.** Residential biomass combustion is a significant source of aerosol particles on regional and global scales influencing climate and human health. The main objective of the current study was to investigate the properties of cloud condensation nuclei (CCN) emitted from biomass burning of solid fuels in different cookstoves mostly of relevance to Sub-Saharan East Africa.

The traditional 3-stone fire (3S) and a rocket stove (RS) were used for combustion of wood logs of sesbania (ses) and casuarina (cas) with birch (bir) used as a reference. A natural draft (ND) and a forced draft (FD) pellet stove were used for combustion of pelletized sesbania and pelletized Swedish softwood (sw) alone or in mixtures with pelletized coffee husk (ch), rice husk (rh) or water hyacinth (wh). The CCN activity and the effective density were measured for particles with mobility diameters of ⌣65, ⌣100 and ⌣200 nm, respectively, and occasionally for 350 nm particles. Particle number size distributions

were measured online with a fast particle analyzer. The chemical composition of the fuel ash was measured by application of standard protocols.

The average particle number size distributions were by number typically dominated by an ultrafine mode, and in most cases a soot mode was centered around a mobility diameter of ⌣150 nm. The CCN activities decreased with increasing particle size for all experiments and ranged in terms of the hygroscopicity parameter, $\kappa$, from ⌣0.1 to ⌣0.8 for the ultrafine mode

and from ⌣0.0 to ⌣0.15 for the soot mode. The CCN activity of the ultrafine mode increased with increasing combustion temperature for a given fuel, and it typically increased with increasing potassium concentration in the investigated fuels. The primary CCN and the estimated particulate matter (PM) emission factors were typically found to increase significantly with increasing potassium concentration in the fuel for a given stove. In order to link CCN emission factors to PM emission factors, knowledge about stove technology, stove operation and the inorganic fuel ash composition is needed. This complicates the use

of ambient PM levels alone for estimation of CCN concentrations in regions dominated by biomass combustion aerosol, with the relation turning even more complex when accounting for atmospheric ageing of the aerosol.





## 1 Introduction

Residential biomass burning comprises a significant source of atmospheric aerosol particles and trace gases on a global scale
(Ludwig et al., 2003). Biomass burning emissions pose a human health risk (Fullerton et al., 2008; Bølling et al., 2009) and
influence atmospheric chemistry (Crutzen and Andreae, 1990) and climate (Bond et al., 2004; Penner et al., 1992). Ludwig et al.
(2003) estimated residential biomass burning to account for about 17% of the total global $CO_2$ emissions in the mid-1990s,
while the absolute emissions have grown significantly in recent decades (Fernandes et al., 2007). The emitted aerosol particles
may scatter or absorb solar/terrestrial radiation directly, and they may also act as cloud condensation nuclei (CCN) and thus
influence climate indirectly through an impact on cloud optical properties and lifetimes (Penner et al., 1992; Albrecht, 1989;
Kaufman and Fraser, 1997). Huang et al. (2018) reported an annually averaged global radiative forcing of -226±5 $mWm^{-2}$
due to the 'warm' cloud indirect climate effects of CCN emitted from residential solid fuel cookstoves. Hence, this global
indirect aerosol cloud climate impact is likely to be significant.

Biomass-fueled cookstove emissions of particulate matter (PM) per mass of combusted dry fuel depend on the stove (com-
bustion conditions) and fuel type used. The PM emissions from the traditional 3-stone (3S) stove have been reported to be
significantly higher those from the rocket-type stoves (Jetter and Kariher, 2009; MacCarty et al., 2010; Just et al., 2013), which
was further pronounced when atmospheric ageing and secondary organic aerosol (SOA) were taken into account (Reece et al.,
2017). Forced draft stoves emit relatively lower amounts of PM per mass unit of combusted dry fuel (Jetter and Kariher, 2009;
MacCarty et al., 2010), and the SOA emissions are also relatively lower (Reece et al., 2017). The particles emitted from res-
idential biomass burning are typically comprised of elemental carbon, organic and inorganic compounds depending on the
composition of the fuel, the combustion conditions and ageing of the emissions (Bølling et al., 2009; Lamberg et al., 2011;
Reece et al., 2017). Among inorganic species, elements such as K, Na, S, Cl and Zn are dominating the fine mode particles
(<2.5 $\mu$m) present mainly as different alkali salts (e.g. KCl and $K_2SO_4$), while more refractory elements like Ca, Mg and Si
preferably are found in the coarse fraction (>2.5 $\mu$m) (Boman et al., 2004; Joeller et al., 2005; Obernberger et al., 2006).

Studies of atmospheric particles sampled over different biomass burning areas in Africa have also shown that the fine
particles emitted from burning of vegetation are dominated by S, Cl, Zn, K and P (Gaudichet et al., 1995). Furthermore,
the major fraction of the Si, Ca and Mn found in the atmospheric particles could be attributed to the burning of vegetation,
although present in the coarse mode. It was also suggested that Ca and Si, together with e.g. Fe, Al and Ti, can be present as
soil-derived particles. Still, in another study on atmospheric particles in smoke plumes from biomass burning, also some fine
mode Ca-bearing particles were present, in parallel with the dominating alkali salts, soot and organic tar balls (Li et al., 2003).
In this paper, the CCN activity will be presented in terms of the hygroscopicity parameter $\kappa$ in a similar fashion as introduced
by Petters and Kreidenweis (2007). It is challenging to assess the CCN properties of biomass burning aerosol emissions in
general due to significant variability with varying fuels and combustion conditions. In addition, ageing is potentially of great
importance. The CCN activity of aerosol particles related to various simulated wild fire emissions has been reported to range
from a $\kappa$=0.04 to $\kappa$=0.8, with a tendency of decreasing $\kappa$ with increasing particle size and often externally mixed particles
(Petters et al., 2009). Engelhart et al. (2012) reported $\kappa$ to range from 0.06 to 0.6 for similar fresh emissions with the $\kappa$





values converging towards 0.2±0.1 due to photochemical ageing and formation of SOA with an average $\kappa_{SOA}$=0.10±0.02. Martin et al. (2013) investigated aged and freshly emitted aerosol from combustion of beech in a residential log wood burner. They reported a CCN activity ranging from an apparent $\kappa$=0.03 to $\kappa$=0.39 depending on the burning phase and decreasing with increasing particle size from 50 to 200 nm. They also reported an increasing CCN activity up to $\kappa \approx$0.16 due to photo-chemical

ageing over some hours for a given particle size with lower initial CCN activity. Photo-chemical ageing of biomass burning aerosol may also result in new particle formation with a potential significant increase in the number concentration of CCN (Hennigan et al., 2012; Engelhart et al., 2012).

Emission factors of CCN for biomass combustion have to our knowledge previously not been studied in detailed experiments. The emission factors of CCN for wild fire biomass combustion have been estimated to be at the order of $0.8 \cdot 10^{15}$ to $1.7 \cdot 10^{15}$

$kg^{-1}$ relative to the dry mass of various fuels and assuming a supersaturation of 0.5% (Andreae, 2019). Mena et al. (2017) modeled the CCN properties of a residential biomass combustion plume and they concluded that coagulation limits the CCN emission factor for a supersaturation of 1.0% to a maximum of $\smile 10^{16}$ per kg of fuel.

The main objective of the Salutary Umeå STudy of Aerosols IN biomass cookstove Emissions (SUSTAINE) campaign was to study sustainable approaches to residential solid biomass combustion in cookstoves relevant for Sub-Saharan East Africa. In

this context, fuel sustainability, combustion conditions and aerosol emissions were studied. The main part of the SUSTAINE campaign was carried out in the fall of 2016, when a wide range of properties related to aerosol emissions were studied under well-controlled laboratory conditions. The four investigated cookstoves represented technologies ranging from very simple (3S) to advanced (forced draft) systems. The tested biomass fuels are either currently in use or potentially more sustainable options relevant to Sub-Saharan East Africa, with softwood pellets used in co-combustion and as reference along with birch

wood logs. The approach allowed us to study how aerosol emissions depend on the stove technology and the fuel. A number of studies with different focus were carried out during the campaign, and the ice nucleating ability of the aerosol has been presented in a separate study (Korhonen et al., 2020).

Numerous previous studies have focused on bulk PM emissions from biomass stoves, whereas few studies have been carried out on the emitted particle number size distributions and the associated CCN properties, which are very important parameters

in a climatic context. The present study focuses on the CCN activity and CCN emission factors for a range of current and potentially future aerosol emissions from household cooking solutions involving biomass combustion. Fresh emissions were generally studied, but the influence of atmospheric ageing was also investigated on a qualitative basis for selected experiments.

## 2  Theory

The CCN activity will be expressed by use of the hygroscopicity parameter $\kappa$. $\kappa$ was introduced by Petters and Kreidenweis

(2007) and is approximated well by:

$$\kappa = \frac{4A^3}{27D_p^3 ln^2(1 + SS_c/100\%)} \qquad with : A = \frac{4\sigma M_{\mathrm{w}}}{\mathrm{R}T\rho_{\mathrm{w}}} \qquad (1)$$





where $SS_c$ is the critical supersaturation in %, $M_w$ is the molar mass of water, $\rho_w$ is the density of water, $\sigma$ is the surface tension, $R = 8.314\,\mathrm{J\,(K\,mol)^{-1}}$ is the universal gas constant, $T$ is the absolute temperature and $D_p$ is the dry particle mobility diameter.

A very useful aspect of the $\kappa$ framework is that the hygroscopicity parameter of internally mixed particles can be estimated by volume weighted addition of the $\kappa$ values of the pure compounds (Petters and Kreidenweis, 2007; Frosch et al., 2011; Kristensen et al., 2014):

$$\kappa_{\mathrm{add}} = \sum \varepsilon_i \kappa_i \tag{2}$$

where $\kappa_i$ is related to species $i$, and $\varepsilon_i$ is the corresponding volume fraction of species $i$ in the dry particles.

The effective particle density ($\rho_{\mathrm{eff}}$) versus particle mobility diameter of soot agglomerates can be modeled by:

$$\rho_{\mathrm{eff}} = \frac{6K \cdot D_p^{\epsilon_m - 3}}{\pi} \tag{3}$$

where $K$ is an empirical constant and $\epsilon_m$ is the mass-mobility exponent. $K$ and $\epsilon_m$ can be determined empirically from least squares fits to measured data as described by Rissler et al. (2013).

## 3   Materials and Methods

**3.1   Stoves and fuels**

Four different stoves and seven different fuels were selected for this study in order to cover a wide range of combustion technologies and associated aerosol emissions. The simple 3S and a more advanced rocket stove (RS) were used for combustion of wood logs of *sesbania sesban* (ses) and *casuarina equisetifolia* (cas) with Swedish birch (*betula pendula*, bir) used for comparison. A natural draft and a forced draft stove were used to combust pelletized biomass of *sesbania sesban*, coffee husk

(ch), rice husk (rh), water hyacinth (wh) and with commercially produced softwood pellets (sw) of a pine/spruce mixture used for comparison. The coffee and rice husk as well as water hyacinth pellets were combusted in 50%-50% mixtures by mass with the softwood pellets in order to ensure comparable combustion conditions for those pelletized fuels.

    The African fuels were all collected in western Kenya in the vicinity of Lake Victoria. *Casuarina equisetifolia* and the faster growing *sesbania sesban* are nitrogen fixing tree species with a density similar to and lower than birch, respectively.

Coffee and rice husk are industrial residues, which currently are treated as waste. Water hyacinth is fast growing and poses an environmental problem related to the eutrophication of Lake Victoria. Hence, application of such fuels in gasifier cookstoves may be relevant from a sustainability and circular economy perspective. Furthermore, the selected fuels represent diverse properties with significantly varying ash content and chemical composition, which is presented and discussed further below in section 4.3.2 in context with the inferred CCN activity. The wood logs had a triangular cross section with dimensions of 2.6

cm times $2.5\pm0.2$ cm, with lengths of 21 and 17 cm for the 3S and the RS, respectively. The pellets were all 3 mm in diameter with a length of 8 mm. The humidity of the pelletized fuels was in the range 7-10%. The humidities of the wood logs of birch,





casuarina and sesbania were about 16, 10 and 14%, respectively. Additional information about the fuels has been presented by Korhonen et al. (2020).

## 3.2 Experimental approach

Several campaigns on biomass combustion have previously been carried out in the Thermochemical Energy Conversion (TEC) laboratory at Umeå University, Sweden. The basic experimental aerosol measurement set up and ageing procedure are very similar to previous studies carried out at the same facility (e.g. Martinsson et al., 2015). The experiments were carried out as a modified version of the standardized water boiling test 4.2.4. In short, a pot containing 5.0 L of water initially at a temperature of 20°C was placed on top of the respective stove and heated up during all experiments to ensure controlled standard cooking

conditions. The combustion was carried out under a hood with a flue-gas fan maintaining a constant flow rate of 2.4 m³/min and ensuring the emitted aerosol being transported through the sampling system. Water vapour from the pot was directed away from the sampling system. Some aerosol measurements were carried out directly in the flue gas, while others including the CCN measurements were carried out on samples emitted into a stainless steel aerosol storage chamber.

The more advanced stoves using pellets were operated with 1.0 kg of fuel and ignition was aided by addition of 12 g of

ethanol. After the initial combustion phase of ∽5 minutes, the intermediate combustion phase was characterised by a constant fuel conversion rate for about 50 minutes before entering the burn out phase. The chamber fillings for the pellet stoves relevant to the current study were generally carried out during the intermediate combustion phase.

The 3S and the RS experiments would typically last until the boiling point of water was reached and boiling maintained through 45 minutes. Also here, the ignition was aided by addition of 12 g of ethanol, which was followed by an intermediate

phase with flaming combustion. Wood logs (about 0.2 kg) were added when the flame intensity decreased or burnout combustion dominated. An experiment would typically include >5 fuel additions. A chamber injection would typically represent roughly one full cycle for a hot stove including (i) fuel addition, (ii) the intermediate phase (flaming combustion) and (iii) burn out combustion. The intermediate combustion phase would typically dominate the time window during injection into the chamber. The aerosol associated with that combustion phase was by number highly dominated by the ultrafine particles. The

particle mode centered near 150 nm, in this study referred to as the soot mode, was typically relatively more pronounced for the burn out phase and the fuel addition.

## 3.3 Instrumentation and measurements

A wide range of online aerosol particle measurements were carried out. Only the instrumentation relevant to the current study will be presented here. The stoves were operated on a scale logging the change in mass of the fuel with a 5 s time resolution.

For selected experiments, aerosol emissions were injected into a 15 m³ stainless steel chamber from which the CCN counter (CCNc) sampled. The injections into the chamber typically lasted for 10-40 minutes depending on the experiment. For the pellet stoves, the injections represented the relatively constant emissions from the bulk of the combustion phase of the experiments. Chamber injections related to the 3S and the RS typically represented an average of different burning phases as described above. The temperature inside the aerosol storage chamber was approximately 20°C and the relative humidity (RH) was about



20-25%. Chamber conditions were intended to be kept constant with as little dilution as possible during experiments. The chamber was flushed with filtered air in between experiments. In addition, prior to selected experiments involving simulated aerosol ageing, the chamber was cleaned with ozone in order to minimize the concentration of potential secondary aerosol precursors not originating from the current stove emission. An overview of the experiments involving chamber fillings and CCN measurements are included in Table 1.

The CCNc (Droplet Measurement Technologies) was operated in flow scan mode (Moore and Nenes, 2009) in a similar fashion as described by Wittbom et al. (2014). The total flow rate ($Q_t$) of the CCNc was controlled during repeated cycles. $Q_t$ increased at a constant rate from 0.2 to 1.0 lpm over 120 s followed by a constant $Q_t$=1.0 lpm for 20 s and a rapid decrease in $Q_t$ down to 0.2 lpm. $Q_t$ was kept constant at 0.2 lpm for 20 s before the cycle was repeated. Only the part of the cycle with increasing $Q_t$ was used in the data analysis. A mass flow controller (MFC) was operated in parallel with the CCNc with the

flow rate continuously adjusted so that the total flow rate of the CCNc and the MFC added up to 1.0 lpm.

The CCNc and the MFC sampled in parallel with an aerosol particle mass analyzer (APM, Kanomax) downstream of a differential mobility analyzer (DMA, TSI-3071), which was operated with a sheath flow rate of 8.0 lpm and a sample flow rate of 2.0 lpm. The voltage of the DMA was systematically alternated between 3-4 set-points for sampling of quasi-monodisperse aerosol particles with mobility diameters of ⌣65, ⌣100, ⌣200 and occasionally ⌣350 nm, respectively. The DMA voltage was

typically kept at a constant set-point for ⌣7 mins, which allowed for obtaining an APM spectrum and 1-2 full CCN flow scan cycles. The DMA and the APM were calibrated during the campaign by use of PSL spheres with nominal diameters of 0.1 and 0.2 $\mu$m, respectively. The APM suffered from minor leaks for the higher rotation speeds during parts of the campaign, which resulted in a limited data set of high quality.

Three different constant temperature gradients (dT) were applied along the CCNc column (4, 10, 18 K) - which allowed for

a coverage of supersaturations ranging from ⌣0.1 to ⌣1.5%. During an experiment, the dT was changed manually in order to obtain reasonable CCN activation spectra for the different particle mobility sizes investigated. Only CCN flow scan cycles with constant dT were used in the data analysis. The supersaturation of the CCNc was calibrated during and after the campaign by use of ammonium-sulfate particles in a similar way as described by Wittbom et al. (2014). The reproducibility of repeated calibration measurements was significantly higher in the current study relative to the variations presented by Wittbom et al.

(2014), which we contribute to our efforts of improving the flow control. Hence, the errors in the supersaturation in the current study are relatively lower by a factor of ⌣2 relative to the calibration results presented by Wittbom et al. (2014).

Particle number size distributions were measured online in the flue gas with a fast particle analyser (DMS 500; Cambustion). The emitted aerosol was diluted by a factor of ⌣20 or more in the hood depending on the aerosol emissions and burn rate. The residence time of the flue gas in the hood and the flue gas channel was at the order of 2 seconds followed by an ejector

dilution step using dry clean air with a ratio of 1:100 for most of the experiments. Thus, two dilution steps were introduced before the fast particle analyser measurements. A scanning mobility particle sizer (SMPS, TSI model 3938) spectrometer was used to measure particle number size distributions in the aerosol storage chamber.

A high-resolution time-of flight soot particle aerosol mass spectrometer (AMS) (DeCarlo et al., 2006; Onasch et al., 2012) was used for chemical characterisation of the submicron aerosol PM. Application of a laser with a wavelength of 1064 nm





inside the AMS allows for vapourisation of soot particles. The AMS was typically sampling from the flue gas line during combustion after a dilution step (ratio 1:100). For the experiments where PM was injected into the aerosol storage chamber, the AMS would after the transient measurements also be used to sample from the chamber.

A potential aerosol mass reactor (PAM) (Kang et al., 2007) was used to simulate intense atmospheric ageing. Two mercury UV-lamps with peak wavelenths of 185 and 254 nm were used to produce hydroxyl radicals (OH) and ozone ($O_3$) inside the

PAM for oxidation of the aerosol sampled from the aerosol storage chamber. The $O_3$ concentration in the PAM was measured with an ozone monitor (Thermo Scientific, model 49i). The OH concentrations inside the PAM were estimated from calibrations with $SO_2$ in the absence of and in the presence of high concentrations of biomass burning aerosol particles (Li et al., 2015). The higher PM loading relevant in the current study only resulted in minor reductions in the estimated OH concentrations. Efforts were made to reduce the background of secondary aerosol particle precursors in the system as described above, and

the background was generally found to be negligible compared to the combustion aerosol signal. Three different levels of photochemical ageing would typically be applied in each experiment ranging from ⌣1 to ⌣10 days of atmospheric ageing assuming an average ambient OH concentration of $1.5 \cdot 10^6$ cm$^{-3}$. The CCNc, APM, SMPS and an aethalometer were used for sampling directly from the aerosol storage chamber, or alternatively in parallel through PAM with a total flow rate of about 5 lpm.

The concentration of $CO_2$ was measured online in the flue gas with a non-dispersive infrared sensor (LI 840A, LI-COR). The concentration of carbon in the fuels was obtained from application of the EN ISO 16948 protocol. This information was used to normalise particle emissions to dry fuel consumption as described in more detail below. The ash content and inorganic elemental concentrations were investigated by use of the EN 14775 and the EN 15289, 15290 and 15297 protocols, respectively.

### 3.4    Data analysis

Throughout this study we will refer to the aerosol particles present in the flue gas and initially injected into the aerosol storage chamber as freshly formed or primary, while particulate matter formed in the flow reactor will be considered secondary aerosol.

Determinations of the critical supersaturation, and estimations of particle mixing state (internal/external) for particle mobility diameters of ⌣65, ⌣100, ⌣200 and occasionally ⌣350 nm were obtained through the following procedure: Relaxed step functions were fitted to the CCN concentration vs total flow rate (supersaturation) in a similar way as done by Wittbom et al.

(2014). Multiply charged particles were accounted for, and the critical supersaturation was obtained for 50% of the singly charged particles having activated into cloud droplets. All CCN spectra were visually inspected to ensure that the full range of the spectra were available for analysis, and to investigate for indications of externally mixed particle populations as reported by Petters et al. (2009).

Droplet growth kinetics were investigated by comparing the average droplet mode size measured by the optical particle

counter (OPC) near the exit of the growth column with ammoniumsulfate calibrations used as reference. Experiments with aerosol particles having the same critical supersaturation and applying the same temperature gradient along the CCNc column were used for the intercomparison.





Emission factors of CCN were obtained by normalisation of particle concentrations in the flue gas to dry fuel consumption by assuming 90% of the carbon in the fuel being converted to $CO_2$ in the flue gas. We applied a very simple model to estimate

the CCN emissions due to the discretized information about the CCN activity at different particle diameters. The $\kappa$ value for a given particle mobility diameter was estimated by a linear inter- and extrapolation relative to the particle diameters for which $\kappa$ was measured. The $\kappa$ value below mobility diameters of ⌣65 nm was assumed constant and identical to that obtained for ⌣65 nm. Likewise, the $\kappa$ values for the particles with mobility diameters larger than ⌣200 nm (or alternatively ⌣350 nm) were assumed constant and equal to the value for 200 nm (or 350 nm if available). Thus, the critical supersaturation for any particle

mobility size was estimated, and in all cases but one, the modeled $SS_c$ increased monotonically with decreasing $D_p$. This allowed for estimating the CCN emissions as a function of supersaturation by integration of the average particle number size distributions measured in the flue gas taking the modeled $SS_c(D_p)$ into account. The ability of an aerosol particle to activate into a cloud droplet depends on the dry particle size and the chemistry (largely 'hygroscopicity') for a given supersaturation (Eq. 1). As mentioned above, we generally found that the particle size dominated over the $\kappa$ value with this simple interpolation

approach with one exception. The exception (RS-cas#2) had a pronounced soot mode with low $\kappa$ dominating near ⌣100 nm, while the ultrafine mode dominating near ⌣65 nm had a significantly higher $\kappa$ value resulting in cloud droplet activation of ultrafine particles at lower $SS_c$ than for the soot particles with larger $D_p$. In that particular case, the CCN emission factors were estimated in a similar fashion, but with a separate integration of the two modes obtained by lognormal fits to the modes. It is unclear, whether a similar issue (a non-monotonic $SS_c(D_p)$ distribution) could have been observed for other experiments,

if a higher size resolution in CCN measurements was provided. In this context, it is worth noting that ambient measurements of the CCN activity based on similar integration of particle number size distributions (assuming particle size dominating over $\kappa$)) (e.g. Herenz et al., 2018) may be biased in a range near two overlapping size modes in case (fresh) biomass combustion emissions or similar dominate the aerosol particle population.

The CCN emission factors for the simulated ageing with PAM were estimated in a somewhat similar fashion. In those cases,

the emission factors were estimated with focus on the soot mode by considering the measured change in CCN activity. The average size of the soot mode and its CCN activity were typically unaffected during storage in the aerosol storage chamber for up to ⌣60 minutes. In contrast, coagulation significantly influenced the primary ultrafine particles, which complicated assessing the emission factors related to the isolated effect of photochemical ageing of those. However, simple $\kappa$ considerations (Eq. 2) show that the main ageing effects on the CCN population can be expected for the soot particles with initial relatively low $\kappa$

values. Ideally, the increase in $D_p$ with ageing should also be taken into account, but we were typically not able to determine that with reasonable accuracy from the particle number size distributions, despite clear indications of a particle growth based on CCNc, APM and AMS measurements. Therefore, the inferred shift towards lower $SS_c$ for the aged soot particles can in this context be considered a lower estimate based on changes in chemistry. In addition, nucleation mode particles generally formed during the simulated ageing, and in a couple of cases, these freshly formed particles grew large enough to play a (minor) role

as CCN. The emission factor of these secondary particles was in one case estimated by relative comparison to the soot mode particle number concentration.





The effective particle density was inferred from Gaussian fits to the particle number concentration versus voltage with results for PSL spheres used as reference. Our approach does not account for multiply charged particles, which may lead to the inferred effective density being biased slightly high (by up to ⌣10%) (Rissler et al., 2013). However, for most of the experiments, we would expect the bias for the ⌣65, ⌣200 and ⌣350 nm particles to be less pronounced due to often relatively small potential fractions of multiply charged particles. The APM data obtained shortly after chamber injections were used in conjunction with the particle number size distributions for estimation of the freshly emitted PM. The effective particle density was linearly interpolated from ⌣65 to ⌣100 nm, and from ⌣100 to ⌣200 nm, respectively. The effective particle density was assumed constant and equal to that at ⌣65 nm for the smaller particles. For the particles larger than 200 nm, the effective density was estimated/modeled in a similar fashion as done for soot particles by Rissler et al. (2013) with fits including 200 and 350 nm particles when available. For experiments where APM data were incomplete or missing, values for $\rho_{\text{eff}}$ from similar experiments were assumed. Emission factors of the particulate matter with a diameter up to 0.5 $\mu$m (PM$_{0.5}$) were estimated from the normalised particle number size distributions measured in the flue gas in conjunction with the obtained size dependent $\rho_{\text{eff}}$.

## 4 Results and Discussion

### 4.1 Particle number size distributions

Some examples of normalised particle number size distributions measured in the flue gas with the fast particle analyser are shown in Fig. 1. The presented particle number size distributions were averaged over the time window during which injection into the aerosol storage chamber was carried out. The distributions have been adjusted for the dilution rates and normalised to the corresponding consumption of dry fuel mass. The size distributions associated with the 3S and the RS were typically averaged over periods with significant variations depending on the combustion phase, while the size distributions associated with the NDS and the FDS stayed close to constant during the injections. In general, the features of the presented particle number size distributions measured with the fast particle analyser corresponded relatively well to the initial emissions into the aerosol storage chamber, which were measured with the SMPS. However, the average size of both the ultrafine and the accumulation (soot) mode were typically slightly larger for the chamber versus the flue gas measurements, and it is unclear to which extent it was due to (i) diffusion particle losses of the smallest particles in sampling lines, and (ii) additional coagulation in sampling lines prior to the chamber injections, or (iii) a minor systematic offset between the instruments.

Remarkable differences can be observed between the particle number size distributions for the freshly emitted aerosol for the different experiments. Some distributions are clearly bimodal, with a mode centered around a mobility diameter of 10-40 nm and another mode centered around ⌣150 nm (e.g., RS-cas, FDS-sw). Below we will present results indicating that the ultrafine modes often appear dominated by inorganic compounds potentially with a significant organic fraction, while the modes centered near a mobility diameter of ⌣150 nm typically are comprised of soot particles dominated by refractory black carbon mixed with varying fractions of organic and inorganic components. Other particle number size distributions appear bimodal with overlapping ultrafine and soot modes (3S-cas, RS-cas, RS-ses, ND-ses). The particle number size distributions





related to the forced draft stove and mixed fuels appear unimodal (FD-sw-rh, FD-sw-ch, FD-sw-wh), but a modest soot mode
       was present in all those cases. We speculate that the easily combustible sw-pellets with higher burn-rate in the FDS resulted in
       quenching of the flame on the bottom of the pot leading to enhanced soot emissions, while the pellet mixtures with relatively
       lower burn-rate were likely to result in relatively lower flame height and no quenching of the flame inside the FDS resulting
       in low soot emissions. In general, the particle number size distributions compare well with previously published results for
comparable experiments (Just et al., 2013; Shen et al., 2017; Mitchell et al., 2019).

       The particle number size distribution presented for RS-ses appears to have the most dominant soot mode in terms of the
       number concentration, and we investigated the conditions for that experiment in more detail. It turned out, that two relatively
       short chamber injections were carried out with a minor break in between - and during both injections a pronounced soot mode
       was captured. An alternative chamber injection time window had typically resulted in a less pronounced - but still significant
soot mode. Hence, the presented particle number size distribution is not representative of the typical average over an entire
       combustion cycle for the RS-ses. Since this experiment was only carried out once, we have included the size distribution (and
       associated results) in the figure(s), but we note that it is not directly comparable to the other presented average particle number
       size distributions (or CCN emission factors), which are more representative of the aerosol population averaged over an entire
       combustion cycle.

**4.2    Mixing state**

       An indication of whether the particles are internally or externally mixed can be obtained from the shape of the CCN activation
       spectra (Petters et al., 2009). It is particularly of interest to investigate the mixing state in size ranges where two different
       modes overlap, which in our case is relevant for some of the CCN spectra obtained near a mobility diameter of 100 nm. All
       the obtained CCN spectra were visually inspected and we did not observe any clear indications of externally mixed particles as
reported by Petters et al. (2009) in several cases for aerosols from simulated wild fires. In many cases, the slopes of the CCN
       activation spectra were very similar to similarly obtained spectra for ammonuimsulfate particles. Our observations may be due
       to (i) internally mixed particles in the size range near 100 nm, (ii) one mode being dominant by number near 100 nm, or (iii)
       comparable CCN activity of the overlapping modes near 100 nm. Regardless of which of the listed options that can explain our
       observations, we were generally able to infer a single CCN activity from the CCN spectra of appropriate quality in terms of
reasonable and stable dT and reasonable CCN counting statistics. However, it should also be mentioned that the size resolution
       obtained with the DMA in the current study was not optimised for identification of the CCN mixing state.

       **4.3    CCN activity of freshly emitted aerosol particles**

       The CCN activities (average $\kappa$ values) for freshly emitted particles are shown in Fig. 2 and also presented in Table 2. The
       results represent as far as possible the first measurements carried out shortly after injection into the aerosol storage chamber.
The errorbars represent $\pm 2$ standard deviations reflecting the variation between these early measurements. Some experiments
       were carried out up to five times (Table 1). In the cases with repeated experiments, the results presented in Fig. 2 include all
       data of reasonable quality, and the same holds for what is presented in Table 2, with the exception of RS-cas. Thus, large





errorbars do not necessarily reflect large uncertainties in single measurements - rather they may reflect variations within a single experiment or for different chamber injections for similar experiments. The latter is particularly pronounced for RS-

cas for $D_p$=100 nm, where two more extreme cases are included in Table 2, and the two corresponding averaged particle number size distributions are included in Fig. 1. In the RS-cas#1 case, the ultrafine mode dominates near $D_p$=100 nm, which is associated with a relatively high $\kappa$ value (0.31), while in the RS-cas#2 case, the mode centered near $D_p$=150 nm dominates the concentration close to $D_p$=100 nm and is associated with a significantly lower $\kappa$ value (0.06). In general, $\kappa$ decreased with increasing mobility diameter for the size range investigated for all the experiments, which also has been observed in other

comparable studies (Martin et al., 2013; Petters et al., 2009). In addition, it is worth noting that the reproducibility of $\kappa$ values for the 3S-cas and the RS-cas experiments typically is quite high (as indicated by the modest errorbars in Fig. 2), at least for the ⌣65 and ⌣200 nm particles. Hence, the experiments with potentially more pronounced differences in manual cookstove operation conditions result in reproducible average CCN activities for the different aerosol modes.

The CCN activities of aerosol particle emissions from combustion of birch, casuarina and sesbania wood logs in the 3S

vs the RS are shown in Fig. 2.a. These results to some extent allow for investigating the influence of the stove and the fuel separately - keeping in mind that the different combustion cycles were not necessarily represented identically between these different experiments. The CCN activity of ⌣65 nm particles is significantly higher for the RS relative to the 3S for any of the three studied fuels. The same trend is to some extent observed for the ⌣100 nm particles (with the exception of birch), while the CCN activities are similar for the ⌣200 nm particles from the two stoves and usage of the same fuel. From Fig. 2.a it can also

be observed that, for the two stoves respectively, there is a decreasing trend in the CCN activity of particles from combustion of sesbania over casuarina to birch. That trend is evident for both of the stoves and for all of the particle sizes investigated. The results presented in Fig. 2.a, clearly indicate that both the stove and the fuel composition are likely to influence the CCN activity at a given particle size, which will be discussed in more detail below.

The CCN activity for different particle mobility diameters and different pelletized fuels combusted in the forced draft stove

are presented in Fig. 2.b. The $\kappa$ values were relatively high for the softwood mixed with coffee husk or water hyacinth for all sizes investigated, with maximum values at the order of 0.6 for the ⌣65 nm particles. The $\kappa$ values for FD-ses for particle diameters near 100 and 200 nm are similar to those observed for RS-ses. The $\kappa$ values appear low (<0.05) with respect to sw, while they are slightly higher (<0.12) for the mixture of softwood and rice husk. In this context it shall be noted that the particle concentrations near $D_p$=65 nm are very low for the freshly emitted aerosol related to FDS-sw and FDS-sw-rh due to

the ultrafine mode being present at smaller sizes. Hence, the associated $\kappa$ values presented in Fig. 2 are only representative of a tiny number fraction of the aerosol. Coagulation in the chamber resulted in growth of the particle modes initially centered around ⌣15 nm (Fig. 1.b). When those modes had grown to dominate near $D_p$=65 nm, $\kappa$ values up to 0.20 and 0.12 for FDS-sw and FDS-sw-rh, respectively, were detected. In the cases where such trends were observed, the associated $\kappa$ values are included in Table 2. It shall also be noted that in some cases the number concentrations of larger particles are low as can be observed

from Fig. 1, so the $\kappa$ values of 200 nm particles from FDS-sw-ch, FDS-sw-rh or FDS-sw-wh relate to a tiny number fraction of the emitted aerosol particles.



Fewer experiments were carried out with the natural draft stove, and the inferred $\kappa$ values are included in Table 2. There seems to be a tendency of relatively lower $\kappa$ values for the NDS-ses and NDS-sw-ch aerosols compared to the FDS-ses and FDS-sw-ch aerosols, respectively, but the data set is too limited to draw any general conclusions in this respect. In Section 4.3.2 below, we discuss how the presented $\kappa$ values (together with supportive results) can be used to provide qualitative information about the chemical composition of the respective aerosol particles.

### 4.3.1 Effective particle density

The available APM data provide complementary supportive information regarding the aerosol particles, as will be discussed in more detail in the following section. Unfortunately, we do not have APM data from all the chamber experiments due to technical issues with the instrument during parts of the campaign. In Fig. 3, selected typical results are presented.

In general, the effective density decreased with increasing particle mobility diameter for all experiments carried out (including experiments not included in Fig. 3). Typically, the effective density of freshly emitted particles with a mobility diameter of $\smile$65 nm would be found in the range 0.9-1.3 gcm$^{-3}$ including all experiments with reasonable data quality. For the freshly emitted 200 nm particles, typical effective densities were found in the range from 0.35 to 0.5 gcm$^{-3}$ for the RS, the FDS and the NDS applying various fuels, while it was $\smile$0.8 gcm$^{-3}$ for the different 3S experiments. The effective density for the particles with a mobility diameter of 350 nm were lower than for 200 nm whenever measured, with values of 0.2-0.3 gcm$^{-3}$ for the FDS, $\smile$0.3 gcm$^{-3}$ for the RS and $\smile$0.4 gcm$^{-3}$ for the 3S. Similar low effective densities have been reported for fractal like black carbon dominated agglomerates from a variety of sources (Rissler et al., 2013). The relatively higher effective density for the freshly emitted 200 and 350 nm soot particles for the 3S is potentially due to (i) a larger fraction of organic compounds/coatings, and/or (ii) more compact black carbon particles. The first suggestion did most likely play a role, judging from the following observations. Sampling through a thermodenuder was carried out a few times during the campaign. Sampling of the 3S-cas aerosol through the thermodenuder with a temperature of 280°C resulted in a significant reduction of the effective density of the 200 nm particles and also towards more black carbon-type aerosol absorption as measured with the aethalometer. Refractory organic primary matter that passes through the thermodenuder may also be present, and it is in line with significantly higher amounts of primary organic matter reported for the 3S relative to the other types of stoves included in our experiments (e.g. Reece et al., 2017).

Atmospheric ageing was simulated by sampling through an oxidation flow reactor as described above. There was a general tendency of increasing effective density with increasing levels of OH and ozone in the flow tube. In four cases, freshly formed secondary aerosol particles dominated by number around a size of 65 nm (Table 5). An effective density close to 1.4 gcm$^{-3}$ was observed in those cases. Similar densities close to 1.4 gcm$^{-3}$ have previously been reported for various secondary organic aerosol particles (Kostenidou et al., 2007; Kuwata et al., 2011; Nakao et al., 2013).

The consistent increase in effective density with increased ageing for all particle sizes investigated can thus be explained by condensation of SOA onto the pre-existing particles with lower initial effective density. That is supported by the AMS measurements, where the relative increase in PM due to the simulated ageing was due to organic species with fragmentation patterns characteristic of SOA particles.





### 4.3.2 Chemical composition and CCN activity

The observed differences in the CCN activity and the effective density reflect significant differences in the chemical composition of the aerosol particles between the different experiments. The ash content in general and concentrations of major ash forming elements in the fuels, together with the combustion conditions, determine the chemical behaviour including volatilisation of fine aerosol forming inorganic species (Bostrom et al., 2011). In the current study, the total ash content in the fuels varied within almost 2 orders of magnitude from ⌣0.3% for sw-pellets and birch over ⌣2.4% for cas and ses to ⌣25% for rh and wh, and the elemental ash composition varied significantly between the different fuels. For all woody fuels (ses, cas, birch and sw) Ca and K are the main ash elements with less but varying content of e.g. Si and P. For two of the pelletized agro-fuels (ch and wh) very high amounts of K were seen in the fuel ash, although with varying content of other elements. For wh, high concentrations of Si, Al, Ca and Cl were also seen, while for ch, K dominated the ash elements together with some Ca in the fuel. For rh, the fuel ash was totally dominated by Si with a minor fraction of K.

It is rather difficult to exactly infer which inorganic compounds that actually will be present in the fine aerosol particles based on the information in Table 3 alone, due to the complex nature of the ash transformation reactions that govern the behaviour of different elements (e.g. volatilization and condensation) during the combustion process. However, considerable experimental experiences exist together with conceptual fundamental thermochemical models (Bostrom et al., 2011), which give a rather solid framework to support the interpretation of the results seen in the present work related to fine inorganic aerosol formation and CCN activity. Based on the state-of-the-art knowledge in this field, it is expected that elements such as K, Na, S, Cl and Zn, as well as some other trace metals like Pb and Cd, to a different degree will be volatilised during combustion. Depending on the different chemical composition of the flue gases, those volatilised species may react further with gaseous components (e.g. $O_2$, $CO_2$, HCl, $SO_2$) and/or condense onto sub-micron particles during cooling of the flue gases. In small scale biomass combustion applications, more refractory species like Ca, Mg and Si are expected to stay in the bottom ashes, as solids or melts, with only a minor fraction forming coarse fly ash particles entrained in the flue gases. Still, elements like Si and P play a vital role in determining the degree of volatilization of alkali (K and Na) since they readily react both with Ca and K/Na, forming different silicates and phosphates, most often found in the residual ash in stoves and boilers (Bostrom et al., 2011).

The measured CCN activity may in some cases provide indirect information about the chemical composition of the aerosol particles. Some available $\kappa$ values of various compounds potentially present in the aerosol particles are listed in Table 4. Different potassium salts have been identified in biomass burning aerosol particles in ambient as well as laboratory studies. Species such as KCl, $K_2SO_4$ and $KNO_3$ have been identified as major inorganic components in atmospheric aerosol particles sampled over or adjacent to areas with wildfires in California (Silva et al., 1999) and Africa (Li et al., 2003; Gaudichet et al., 1995). Rissler et al. (2005) reported KCl, $K_2SO_4$ and in some cases $K_2CO_3$ to be the dominant species in fine particulate matter emitted from combustion of moist forest residue and stem-wood pellets in district heating boilers. Furthermore, the detailed composition and morphology of sub-micron (single) particles emitted from small-scale biomass combustion was nicely explored by Torvela et al. (2014), where both soot and inorganic particles where characterised by TEM-EDX. For the inorganic particles, a smaller nucleus of ZnO was in many cases seen in the core of single alkali salt particles (mainly KCl and



$K_2SO_4$). Those potassium salts of potential relevance to the particle phase are included in Table 4 with relatively high $\kappa$ values (0.5-1).

     Secondary organic aerosol particles typically have a $\kappa$ value of 0.1 as found in the current study (presented in Section 4.6 below) and in a previous study on emissions from biomass combustion (Engelhart et al., 2012). Engelhart et al. (2012) estimated primary organic aerosol (POA) from biomass combustion to have a $\kappa$ of 0.09, but it could in general potentially be

lower due to the potential presence of relatively large molecules and not necessarily very oxidised species. Freshly emitted and uncoated soot particles generally have very low $\kappa$ values close to 0 (e.g. Wittbom et al., 2014).

     It is not possible to include all relevant species in Table 4 due to complex chemical matrices and/or unavailable $\kappa$ values. A few Ca or Si species are included in Table 4, typically with low $\kappa$ values. The significant differences in $\kappa$ values between different relevant species allow for extracting information about the aerosol chemical composition on a qualitative level, which

will be discussed in more detail below. The link between the inferred $\kappa$ values and the associated chemical composition for internally mixed particles is based on $\kappa$ addition (Eq. 2). In short, particles with $\kappa$ values close to 0 can be explained by the dominance of elemental carbon and/or organics/inorganics with low solubility/hygroscopicity. On the other hand, particles with $\kappa$ values of >0.5 are likely to be dominated by inorganic soluble salts, potentially the potassium species listed in Table 4. Particles with intermediate $\kappa$ values may be comprised of internally mixed particles of species with different $\kappa$ values and/or

dominance of organic species in case $\kappa$ is in the neighborhood of 0.1. The interpretation can be supported by the measured effective density to estimate whether low $\kappa$ particles can be explained by soot agglomerates or low solubility organic/inorganic species.

     It is interesting to compare the RS-bir, RS-cas and RS-ses experiments. The birch fuel contains significantly less ash, while the total ash fraction is similar for cas and ses. The only significant difference between the ash composition of cas and ses is the mass ratio between the dominating ash elements K and Ca being ᴖ1:2 and ᴖ2:1, respectively. Hence, ses contains the

most easily volatilisable ash mass followed by cas with significantly less in birch. That corresponds very well to the significant differences in the normalized magnitude and size range (related to the particle mass) of the ultrafine modes of the particle number size distributions of RS-bir, RS-cas and RS-ses presented in Fig. 1.a. The relatively high $\kappa$ value of ᴖ0.7 for the ᴖ65 nm particles for RS-ses corresponds well with a dominance of potassium salts. The lower $\kappa$ values of ᴖ0.35 and ᴖ0.2 for

RS-cas and RS-bir, respectively, indicate that other species than potassium salts are present. The most likely explanation would be a significant and a dominant organic volume fraction internally mixed with potassium salts in the ultrafine aerosol particles for RS-cas and RS-bir, respectively. Alternatively, the presence of soot particles and/or various Ca salts in the ultrafine particles would contribute to low $\kappa$ values. However, the soot modes we identified from the particle number size distributions (Fig. 1) did typically not contribute significantly by number near a mobility diameter of ᴖ65 nm. In addition, we find it unlikely with

significant concentrations of Ca salts in the ultrafine particles, which we base on previous laboratory studies introduced above.

     The CCN activity for the ᴖ65 nm particles was significantly lower for the 3S relative to the RS for any of the fuels. That may be due to (i) a relatively lower combustion temperature for the 3S resulting in a lower fraction of inorganic compounds being volatilised (assuming organics to be present in any case), and/or (ii) a relatively larger fraction of inefficiently combusted organic species being present in the ultrafine particles. Judging from Fig. 1.a the particulate mass relevant for the ultrafine





modes appears to be higher for the 3S versus the RS for bir and cas (not including the RS-ses, which may be biased in this context as described above). Those observations of apparently more ultrafine mass in combination with lower $\kappa$ values for the 3S versus the RS indicate that the 3S emits relatively more organic particulate matter in the ultrafine size range of the primary aerosol. Whether the other effect suggested above also plays a role (lower absolute emissions of inorganics for the 3S versus the RS) is not something we can conclude on based solely on these observations, but we would expect such an effect due to the

difference in combustion temperature.

The combustion with the FDS is more efficient and well-controlled, which provides a more reproducible basis for intercomparison of the relation between fuel composition and aerosol properties. For the ch and wh ash, potassium is the dominant element (Table 3) and the corresponding $\kappa$ values for $\smile$65 nm particles are relatively high (0.5-0.6), which can be explained by a significant fraction of potassium salts in the ultrafine particles. The rh ash is very much dominated by Si with a minor

fraction of K, and the corresponding $\kappa$ values for the ultrafine mode is estimated to $\smile$0.12 (measured when the mode had grown by coagulation to dominate at $D_p$=65 nm). Hence, that ultrafine mode is unlikely to be dominated by potassium salts - and dominance of organic species can explain the observation. It is unlikely that soot particles could affect the $\kappa$ value, since there is no indication of significant soot concentrations. Alternatively, the presence of Si salts could potentially contribute to the relatively low $\kappa$ value, however, we would not expect any presence of Si in the ultrafine particles, as discussed above.

For the stoves and fuels investigated, our results indicate that a higher K concentration in the fuel results in a higher CCN activity of the emitted ultrafine particles - which is more pronounced for combustion at higher temperatures (e.g. RS vs 3S). The variations in $\kappa$ values for 200 nm particles ranging from $\smile$0.0 to $\smile$0.2 correlate with the variations in the ultrafine $\kappa$ values for the FDS experiments. However, it should be kept in mind that the particle number concentrations around that particle mobility size in some cases are very low. There is a similar tendency for the 3S and RS experiments - but it is noteworthy that

no significant differences are observed in $\kappa(D_p$=200 nm) between the 3S and the RS for the same fuel. We would expect a relatively larger organic fraction in the larger size range for the 3S relative to the RS experiments as supported by the APM results, so the apparently similar $\kappa$ values may not necessarily reflect similar particles around a mobility diameter of 200 nm - but could potentially be due to different ratios between elemental carbon (EC), organic and inorganic species. In other words, the 3S soot particles are likely to have a relatively higher organic to EC ratio, while the RS soot particles are likely to have a

slightly higher inorganic to EC ratio for a given fuel.

### 4.4 Droplet growth kinetics

The size of droplets formed inside the CCNc was detected by an optical particle counter (OPC). Insoluble organic compounds and mineral dust have previously been shown to delay droplet growth (Asa-Awuku et al., 2009; Kumar et al., 2009), which is the motivation for investigating whether the same may be the case for biomass burning aerosol particles. In the present

study, the size of the formed droplets was systematically investigated. No significant differences were observed between the sizes of the droplets formed on the studied biomass burning aerosol versus ammoniumsulfate particles - when the CCNc operation conditions were identical and the critical supersaturation of the seed aerosol particles similar. Atmospheric ageing of the biomass burning aerosol particles with the lowest CCN activity ($\kappa$<0.1) is likely to result in an increase in the CCN activity,


which is discussed in more detail below. Hence, our results indicate that reduced cloud droplet growth rates are unlikely to
play a role with respect to the studied biomass burning emissions from the investigated fuels and combustion conditions at any
stage of their atmospheric lifetime.

## 4.5   Primary CCN emission factors

Estimated cumulative primary CCN emission factors ($EF_{CCN}$) are shown in Fig. 4. These averaged emission factors are
normalised with respect to dry fuel consumption during the studied chamber injections. For the 3S and the RS, the chamber
injections are typically averaged over different burning phases - and thus a higher variability is expected for those experiments
compared to the pellet stove experiments. In general, the fresh emissions of CCN for a supersaturation of 0.1% are relatively
modest, while for higher supersaturations huge variations are observed for different stoves and different fuels.

In Fig. 4.a, the emission factors for sesbania pellets (NDS) and sesbania wood logs (3S, RS) are included. The CCN emission
factors for the RS were significantly higher relative to the other stoves, which to some extent is likely to be due to the special
conditions for the chamber injections for the RS-ses experiment. Hence, we do not consider the aerosol emission factors for
the RS-ses directly comparable to the other experiments. Fresh CCN emission factors for the RS and birch and two different
RS and casuarina experiments are also included in Fig. 4.a. There is a significant difference in the CCN activity for 100 nm
particles between the two RS-cas experiments, which largely explains the difference in $EF_{CCN}$ observed in the neighborhood
around a supersaturation of 0.3%. This difference is due to different relative representations of different burning phases during
the time period of injection into the chamber. However, the variability in fresh CCN emission factors between the different
RS-cas experiments was very modest relative to the differences to the $EF_{CCN}$ for the RS-bir experiment.

For the RS, the CCN emission factors for sesbania and casuarina are significantly higher than those for birch. The difference
in emitted CCN for the RS-bir relative to the RS-cas and RS-ses experiments exceeds one order of magnitude for a significant
range of intermediate supersaturations. The differences between CCN emission factors for the same fuel and different stoves
appear to be modest compared to the more pronounced differences observed between the different fuels studied. We ascribe
the somewhat similar CCN emission factors for the 3S and RS applying the same fuel to the relatively higher CCN activity ($\kappa$)
for the RS to be compensated (to some extent) by more organic particulate matter and on average larger particles emitted from
the 3S.

In Fig. 4.b, primary CCN emission factors vs supersaturation for the FDS and different fuels are shown, with the NDS and
sesbania pellets included for comparison. These fresh CCN emission factors vary by several orders of magnitude depending
on the chemical composition of the fuel. The fuels containing a significant fraction of potassium (ch, wh, ses) led to a larger
ultrafine particle mode (Fig. 1.b) and also higher $\kappa$ values, and both of those effects impact the estimated CCN emission
factors. The step in the FDS-sw curve near SS=0.7% is due to activation of the entire soot mode within a narrow range of
supersaturations. That soot mode is probably produced due to quenching of the flame on the bottom of the pot, and a more
optimal fuel load can potentially significantly reduce those soot particle emissions.

The CCN emission factors presented in Fig. 4 appear in general to be highly dependent on the inorganic content and compo-
sition of the fuels. The characteristics of the experimental setup related to wall losses and coagulation (dilution rates) may to





some extent influence the inferred CCN emission factors on a quantitative level. A higher degree of coagulation will result in fewer but larger CCN activating at lower supersaturations. However, we consider it highly likely that the qualitative differences

between the inferred CCN emission factors will remain consistent for other sampling systems with different wall losses and co-agulation effects. Limitations with respect to simulating atmospherically relevant emission factors are discussed in more detail in the following section. In general, we conclude, that the inorganic ash content and composition are very likely to significantly influence the CCN emission factors, with a high fraction of potassium in the fuel resulting in high emissions of CCN. In this context, the stove seems to be of less importance within the investigated conditions.

**4.6 Influence of atmospheric ageing on the CCN properties and emission factors**

The simulated atmospheric ageing resulted in formation of secondary aerosol matter over the entire aerosol population as indicated by the CCN and $\rho_{eff}$ results presented in Table 5. Typically, a mode of freshly formed particles (nucleation mode) appeared in the particle number size distribution. In a few cases, the freshly formed particles grew large enough to dominate the particle number concentration in a neighborhood around a mobility diameter of 65 nm. Those cases allowed us to determine

the $\kappa$ values of the freshly formed particles to 0.11-0.14 and 0.10-0.12 for the 3S-cas and NDS-sw experiments, respectively. Engelhart et al. (2012) similarly reported an average $\kappa_{SOA}$=0.10 with a standard deviation of 0.02 for a range of simulated wildfire biomass burning emissions and simulated atmospheric ageing. Our observations are also similar to $\kappa$ values reported for a wide range of SOA particles (Lambe et al., 2011). Hence, the observed $\kappa$ values support that the formed secondary aerosol was highly dominated by organic species as also strongly indicated by the measurements of effective density (Table 5)

and chemical composition discussed above.

The CCN emission factors may increase due to atmospheric ageing. Four effects may cause such an increase. Condensation of secondary aerosol on pre-existing particles will cause (i) an increase in dry particle size and (ii) potentially an increase in $\kappa$. In addition, (iii) oxidation of soot and/or organic species may enhance their CCN activity. Those three effects will shift the CCN population to become active at relatively lower supersaturation, while (iv) new particle formation may increase the

number of CCN, most likely in the high supersaturation range. Changes in $\kappa$ values for the soot mode due to the simulated ageing are presented in Table 5. An increase in $\kappa$ from 0.006 to 0.012 (3S-bir) results in a decrease of the $SS_c$ for 200 nm particles from 0.54% to 0.38% (assuming a temperature similar to that of the upper CCNc column). Similarly, an increase in $\kappa$ from 0.013 to 0.037 (RS-cas#4) for 200 nm particles leads to a decrease in the $SS_c$ from 0.36% to 0.22%, while an increase in $\kappa$ from 0.071 to 0.094 (RS-cas#5) leads to a decrease in the $SS_c$ from 0.16% to 0.14% for 200 nm particles. These examples

illustrate that the effect of atmospheric ageing on the CCN properties are more pronounced when the initial $\kappa$ is significantly lower than that of the SOA of about 0.1.

We mainly ascribe these changes in $\kappa$ to the process (ii) listed above due to the combined increase in $\kappa$ and effective density (Table 5) often associated with little detectable growth in the average soot mode size with the SMPS. In addition, we cannot exclude a minor potential contribution from process (iii). These examples illustrate that atmospheric ageing may increase the

emitted concentrations of CCN, depending on the supersaturation the aerosol particles may get exposed to. Basic $\kappa$ modeling





involving SOA condensation onto emitted particles using Eq. 1 and 2 supports that ageing has a stronger impact on the CCN activity, when the initial $\kappa$ value is significantly lower than that of SOA, which we often observed for the soot mode.

Examples of estimated CCN emission factors for the aged versus the primary aerosol are shown in Fig. 5. The presented aged aerosol emission factors are based on the measured change in $\kappa$ of the soot mode, while growth of the primary ultrafine 560 mode is expected to have significantly less impact due to the relatively higher initial $\kappa$ values. In one case, the effect of freshly formed secondary aerosol CCN is also shown (3S-cas), and this mainly affects the CCN concentrations at the very highest supersaturations included.

Our results indicate that the partitioning of the emitted gas-phase secondary particle precursers between (i) the gas-phase, (ii) formation and growth of freshly formed particles, and (iii) condensation on different pre-existing particle modes may be of great 565 importance for the total CCN emission factors relevant for real atmospheric conditions. The partitioning of the condensable organic matter is likely to depend upon coagulation (dilution/dispersion), the surrounding ambient aerosol particle population and gas-phase constituents (polluted versus pristine environment) including available oxidants (e.g. solar radiation intensity and daytime vs nighttime chemistry). It was not possible to estimate the relative importance of such complex parameters in the current study, but the effects are discussed in more detail below.

It is challenging to simulate atmospheric ageing of the studied aerosol on a quantitative level. Prior to our flow reactor experiments, the aerosol is transported through a hood and some tubing, which may be representative of the geometry of a chimney, where wall-losses and coagulation significantly may influence the aerosol particle population. However, in our experimental setup, significant dilution is introduced at an early stage making our conditions more representative of open ambient combustion emissions with pronounced aerosol dispersion. Hence, the studied freshly emitted aerosol may be representative of 575 such fresh combustion aerosol emissions in African regions where biomass cookstoves dominate - often operated in the open. Nevertheless, before entering the flow reactor, the aerosol is typically stored for 30-60 minutes inside the aerosol storage chamber, where particle coagulation in some cases occurs, and where wall losses of ultrafine particles and potentially gases may influence the outcome of the experiment. Inside the flow reactor, the aerosol is exposed to unrealisticly high concentrations of OH and $O_3$ for atmospheric conditions - in order to simulate up to several days of ageing. The high concentrations of oxidants 580 are likely to influence the partitioning of the potential secondary aerosol particle precursors. In addition, wall losses of oxidised gas-phase constituents in the flow reactor may bias the secondary aerosol particulate matter yield low. Furthermore, residential cooking often involves more than just boiling water. Hence, actual cooking will typically result in additional emissions of primary particles and gas-phase constituents potentially relevant for additional formation of secondary aerosol (Zhao et al., 2007). Hence, our estimated total CCN emission factors for the aged aerosol are likely to be biased low relative to expected 585 field observations, at least for the relatively low supersaturations. However, on a qualitative level, they do provide insight into how atmospheric ageing influences the CCN properties.

## 4.7 CCN versus PM emission factors

The primary PM emission factors from cookstoves have been reported in many previous studies, and ambient PM measurements are available in many regions dominated by residential biomass combustion emissions. The CCN emissions from biomass



combustion in laboratory and field settings are generally scarce, and it is of high relevance to investigate how CCN emissions relate to PM emissions. The estimated primary CCN versus estimated primary $PM_{0.5}$ emission factors for the different stoves and fuels are shown for supersaturations of 1.0% and 0.5% in Fig. 6.a and Fig. 6.b), respectively. Those two supersaturations were chosen to represent a high supersaturation of relevance to highly convective clouds, and a supersaturation level where the soot mode in many cases has activated, respectively. Before interpreting the relation between emitted CCN and PM presented

in Fig. 6, we find it appropriate to discuss how the PM EFs compare to previous findings, as well as potential errors and biases related to the PM.

The estimated primary $PM_{0.5}$ EFs for the forced draft stove range from 0.2 to 0.8 $g/kg_{dryfuel}$ with the exception of the FDS-sw-rh with a very low PM EF of about 0.02 $g/kg_{dryfuel}$. $PM_{2.5}$ EFs have for comparable stoves and full water boiling tests been reported to be about 0.4 $g/kg_{dryfuel}$ (Jetter et al., 2012; Champion and Grieshop, 2019) with a significant fraction

of the PM emissions being associated with the ignition, refueling and burnout phases (Champion and Grieshop, 2019), which have been excluded from our experiments. Hence, most of our estimated $PM_{0.5}$ EFs for the FDS compare well to previous $PM_{2.5}$ EF observations. We note that our $PM_{0.5}$ EF for the FDS-sw is biased high due to non-ideal stove operation.

The estimated primary $PM_{0.5}$ EFs for the RS (0.10-1.47 $g/kg_{dryfuel}$) and the 3S (0.27-1.44 $g/kg_{dryfuel}$) are somewhat lower than values reported for $PM_{2.5}$ in previous water boiling test experiments. The lower reported $PM_{2.5}$ EFs for full water

boiling tests were about 1.4 and 1.8 $g/kg_{dryfuel}$) for rocket stoves and the 3S, respectively (Jetter et al., 2012). We ascribe the differences to be due to (i) exclusion of the ignition and end-of-experiment burnout phases in our approach, and (ii) to some extent PM emissions potentially being present in the 0.5 to 2.5 $\mu$m size range (Just et al., 2013).

It is worth noting that the results shown in Fig. 6 are specific to the experimental setup and procedure, and they are associated with random errors and biases. However, the qualitative and relative comparison between estimated PM and CCN emission

factors is in many respects robust in the sense that both parameters are inferred from the same particle number size distribution. Hence, biases and random errors associated with those measurements would in many respects cancel out when comparing the CCN and PM emission factors, with the latter being more sensitive to the very largest particles.

There are three noteworthy outliers in Fig. 6, the FDS-sw-rh, FDS-sw and RS-ses, which are of relevance to discuss before interpretation of the more general features. As discussed above, the FDS-sw experiment produced a pronounced soot mode,

which we ascribe to quenching of the flame on the bottom of the pot. Similar experiments carried out at a later stage confirmed that it was possible to operate the FDS-sw without production of a significant soot mode. Optimal operation would lead to a significant reduction in both CCN and PM EFs. The RS-ses experiment represents a chamber filling where the soot mode was over-represented, as discussed above. We carried out a few simple sensitivity tests applying the same $\kappa$ and $\rho_{eff}$ distributions, but with different chamber injection windows representing a full combustion cycle. Those simple tests indicate that the $PM_{0.5}$

may be biased high by up to 0.5 $g/kg_{dryfuel}$) relative to the other RS and 3S experiments, while the alternatively estimated CCN EFs remained within ±25% of the datapoints included in Fig. 6. Finally, the CCN and PM emissions are very low for the FDS-sw-rh experiment despite a significant potassium content in the rh pellets. The inorganic ash of the rh is highly dominated by silicon (Table 3), which may inhibit the aerosolisation of the potassium species, as discussed above (Bostrom et al., 2011).





These three outliers exemplify that the studied emissions can be highly sensitive to the stove operation, the experimental
approach and the composition of the fuel, which is discussed in more detail below.

Several interesting qualitative features can be observed in Fig. 6. First we focus on the 3S and RS results. If we take into
account that the RS-ses PM EFs is likely biased high, then the PM EFs of the 3S are higher than for the RS applying the same
fuel, which is in line with several previous studies (e.g. MacCarty et al., 2010; Jetter et al., 2012). However, the reduction in
PM EFs by replacing the 3S with RS does not appear to influence the primary CCN EFs significantly, which seems to be the
case for the full range of relevant supersaturations (Fig. 4). The reduction in PM EFs going from the 3S to the RS is likely due
to a significant reduction in OC emissions with a higher combustion temperature, which also is likely to enhance emissions of
the more hygroscopic inorganic species. Those two effects largely appear to cancel out with respect to the emitted CCN, for
which, a significantly lower organic fraction in the soot mode may be compensated by a relatively small addition of inorganic
hygroscopic species.

Another interesting feature related to the 3S and RS results is a clear trend of increased CCN and PM EFs with increasing
concentration of potassium in the dry fuel. As discussed above for the FDS-sw-rh, the emissions may not depend on the absolute
potassium concentration in the fuel alone, so it would be highly interesting to further investigate how robust the observed trend
is. Our findings indicate that it may be possible to reduce PM and particle number emissions very significantly by applying
a low-potassium content fuel, which is worth considering from a health perspective. Nevertheless, our observations are not
suited for a full assessment of such potential health benefits, since we have not included ignition and end-of experiment burn-
out phases. In addition, we have no quantitative measurements of the potential contribution of secondary aerosol emissions.
If atmospheric ageing is considered, we would expect the PM EFs of the 3S to increase relatively more than it would for
the RS applying the same fuel (Reece et al., 2017), but it is less clear to which extent the secondary aerosol formation from
atmospheric ageing may be influenced by the choice of fuel.

The PM EFs can be reduced significantly when replacing the NDS with the FDS applying the same fuel assuming optimal
stove operation, which is in line with previously reported results (e.g. Jetter et al., 2012). The PM EFs reduction is likely due
to (i) reduced OC and/or (ii) EC emissions. However, again it is worth noting that this reduction in PM emissions may not be
associated with a reduction in CCN emissions. The FDS experiments applying sw-ch, sw-wh and ses are all associated with
modest PM EFs and some of the highest CCN EFs which again illustrates that improved stove technology may not necessarily
reduce the primary CCN emissions. However, with the very low FDS-sw-rh CCN and PM EFs in mind, it seems like the
inorganic composition of the fuel can be of great importance. There is room for further studies in that context considering fuel
efficiency, availability and sustainability.

In general, it would be useful if the CCN EFs could be directly linked to the PM EFs. It may be possible to provide an
empirical relation between CCN and PM EFs for the 3S or the RS depending on the inorganic content of the fuel, but more
experiments are needed to fully shed light on that. However, it seems impossible to link a given single measurement of PM
originating from cookstove emissions to a given CCN concentration with high certainty unless further physico-chemical aerosol
properties are provided. For example, similar PM emissions can be observed for the FD-sw-wh and ND-sw in Fig. 6.b with a
difference in CCN emission factors of 4 orders of magnitude for an SS=0.5%. However, in those two cases, we would expect





the chemical composition of the PM to differ significantly (pronounced potassium levels versus dominance of EC and possibly

OC, respectively), so information about the chemical composition of the PM (or the fuel) is likely to significantly improve the

ability to estimate the CCN population from the emitted PM. Our results indicate that potassium is likely to play a key role in

that context.

There are a few additional implications and perspectives related to our observations. It seems highly likely that CCN emission

factors from wildfires also will depend highly on the potassium content in the fuel - and potentially the content of other

inorganic aerosol forming elements. So we speculate that significant spatial differences can be expected for wildfire CCN EFs

depending on the chemical composition of the biomass burning.

When assessing the health impact of aerosol particles, it is of relevance to know whether the particles grow hygroscopically

inside the respiratory tract with a high RH level, as it will influence particle deposition and dilution (Löndahl et al., 2007, 2008).

Our observations indicate that the ultrafine particles with moderate to high $\kappa$ values are likely to grow hygroscopically in the

respiratory tract as previously shown (Löndahl et al., 2008) for biomass combustion aerosol, whilst there may be soot particles

with the very lowest $\kappa$ values, which may not show significant hygroscopic growth in the respiratory tract. Our observations

indicate that the content of potassium in the fuel as well as combustion temperature will influence the hygroscopicity of the

soot particles. However, further hygroscopicity studies at relevant RHs are needed for any firm conclusions in that respect.

## 5    Conclusions

The CCN properties of aerosol emissions from various combinations of 4 different cookstoves and 7 different solid biomass

fuels have been investigated. The average particle number size distributions were by number dominated by an ultrafine mode

in all cases and a varying soot mode was present and centered near a mobility diameter of ⌣150 nm.

The CCN activity ($\kappa$) and the particle effective density both decreased with increasing particle size for any of the primary

aerosol emissions studied. For the ultrafine mode, the $\kappa$ values ranged from ⌣0.1 to ⌣0.8, and for the soot mode, $\kappa$ ranged

from ⌣0.001 to ⌣0.15.

The aerosol properties and CCN activity varied significantly depending on the fuel and the stove. There was a tendency of

higher $\kappa$ with improved stove technology and increasing combustion temperature for the ultrafine particles from combustion

of birch, casuarina and sesbania wood logs. This is most likely due to a higher inorganic to organic fraction in the ultrafine

aerosol particles for higher combustion temperature. An increase in combustion temperature most likely reduced the absolute

emissions of primary organic aerosol, while higher temperatures also were likely to enhance the absolute emissions of alkali

salts. In general, higher potassium ash concentration was associated with higher $\kappa$ values for the studied fuels.

The estimated primary CCN emission factors were found to vary substantially depending on the fuel composition, while the

stove appeared to have a less pronounced influence. Simulated atmospheric ageing led to formation of secondary aerosol most

likely dominated by organic compounds. The secondary aerosol condensing onto the soot particles increased the concentration

of CCN for low supersaturations mainly due to an increase in $\kappa$. In addition, new particle formation and growth potentially



increased the CCN concentration for relatively high supersaturations. The secondary aerosol mode had a an effective density
of about 1.4 gcm$^{-3}$ and a $\kappa$ of about 0.1, which are typical values for secondary organic aerosol.

Primary PM$_{0.5}$ emission factors were estimated and found to increase with increasing potassium content for the rocket stove
and the 3 stone stove. The estimated PM$_{0.5}$ emission factors typically decreased with improved stove technology (increasing
combustion temperature) for a given fuel, while the primary CCN emission factors appeared relatively unaffected by improved
stove technology. The reduction in PM emissions with improved stove technology was mainly associated with reduced emis-
sions of organic and elemental carbon. From a CCN emission perspective, the reduced PM emissions were compensated by
elevated emissions of more hygroscopic alkali salts for a given fuel. A given PM emission level can be associated with orders of
magnitude difference in CCN emission factors for a given supersaturation depending on the fuel and the stove. Hence, it appears
challenging to parameterize CCN emissions from the PM emissions from cookstoves without more detailed physico-chemical
information about the aerosol particles. Our results indicate that it is critical to know about the inorganic fuel composition in
order to estimate properties related to the emitted aerosol population and associated CCN properties.

*Data availability.*  All presented data can be requested from the corresponding author T. B. Kristensen

*Author contributions.*  All authors contributed to the experimental work with Kristensen being in charge of the CCN measurements. The data
analysis and manuscript writing were mainly carried out by Kristensen and Falk with contributions from the co-authors.

*Competing interests.*  The authors declare no conflict of interest.

*Acknowledgements.*  This study was supported by the Swedish research councils Formas (grants: 2013-1023, 2015-992 and 942-2015-1385)
and 'Vetenskapsrådet' (grants: 2017-05016, 2013-5021). Also, we acknowledge support from the strategic research area MERGE at Lund
University. R.L. Carvalho acknowledges funding from the Kempe Foundation (grant number JCK-1516) and FCT/MCTES for the financial
support to CESAM (UIDP/50017/2020+UIDB/50017/2020) through national funds. We thank Andrew Grieshop, Erik S. Thomson, Pontus
Roldin and Roland Schrödner for scientific discussions and input.





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



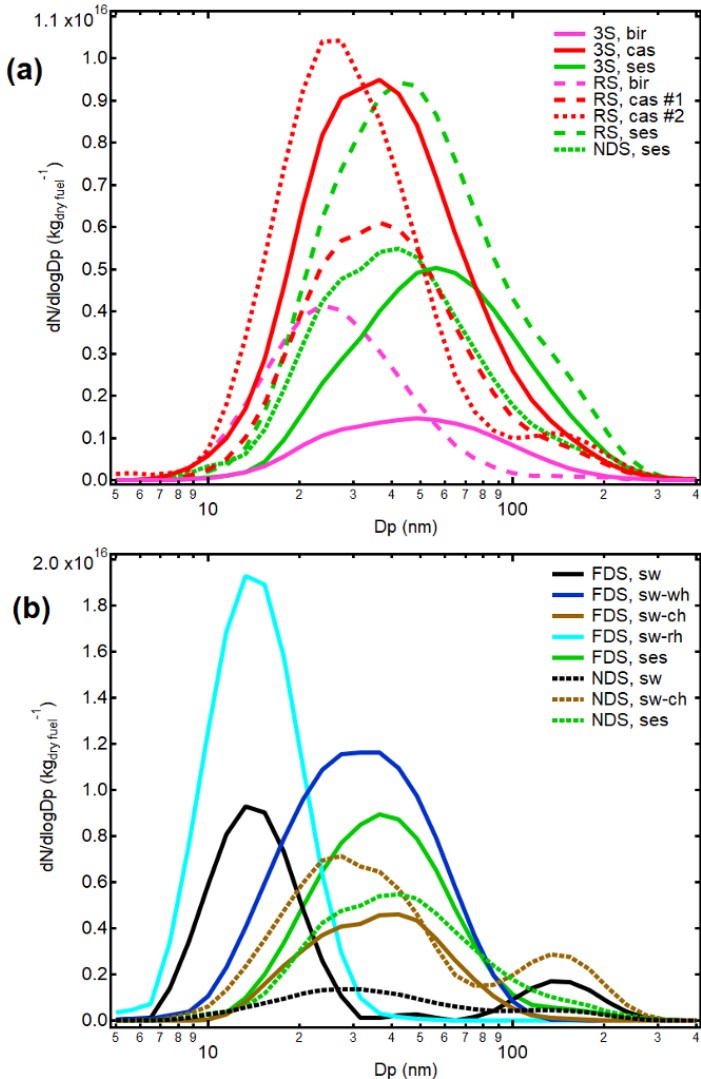

**Figure 1.** Examples of averaged particle number size distributions in the flue gas measured with the fast particle analyser normalised with respect to the dry fuel consumption. The distributions represent averages during the time periods when the respective chamber fillings were carried out. The size distributions in a) from the 3S and RS showed pronounced variability during chamber injections. The size distributions in b) from the FDS and the NDS representing the intermediate combustion phase were typically rather constant during chamber injections. Note that the scale on the ordinate axes are different, and NDS-ses is included in both a) and b) for comparison. The abbreviations in the legend are defined in the caption of Table 1



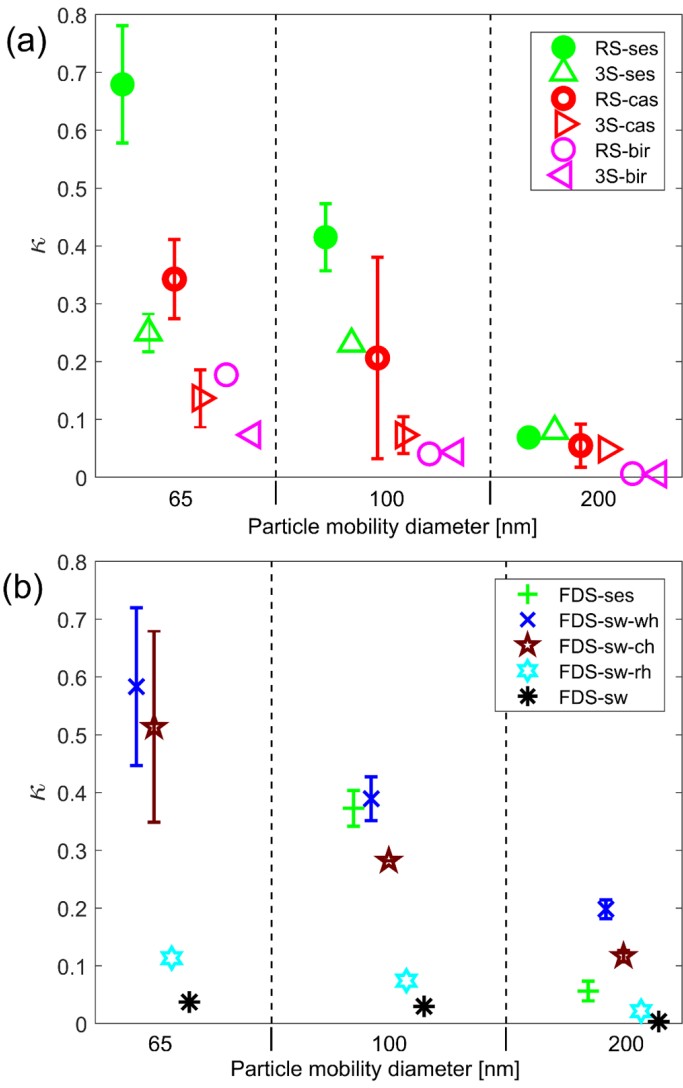

**Figure 2.** The CCN activity expressed as the hygroscopicity parameter $\kappa$ for different fuels and three different particle mobility diameters of $\sim$65, $\sim$100 and $\sim$200 nm, respectively. In (a), for the rocket stove (RS) and the 3-stone (3S) stove, in (b), for the forced draft pellet stove. The data points represent mean values of $\kappa$, while the error bars represent $\pm 2$ standard deviations of the variability between different measurements of the aerosol shortly after chamber fillings. In cases where no errorbars are depicted - they are subsumed by the data points.

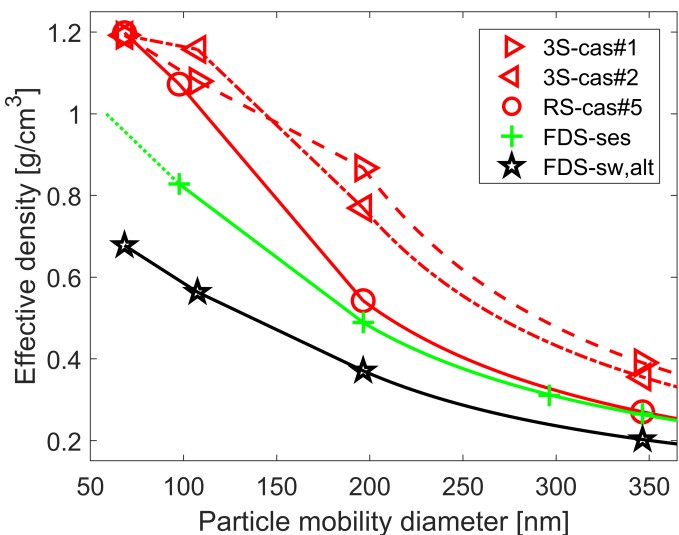

**Figure 3.** Examples of the measured effective densities ($\rho_{eff}$) for primary particles when measurements for $D_p$=350 nm were included. The lines are linearly interpolated for $D_p$<200 nm, while fits to Eq. 3 are applied for $D_p$>200 nm. A datapoint typically represents the average of 2-4 independent measurements from the same chamber filling. The shown FDS-sw,alt results are from an experiment with altered pot-height (Korhonen et al., 2020), and they do originate from the regular FDS-sw experiment, which is presented elsewhere in this study. Examples of additional primary and aged $\rho_{eff}$ for 200 nm particles are included in Table 5.

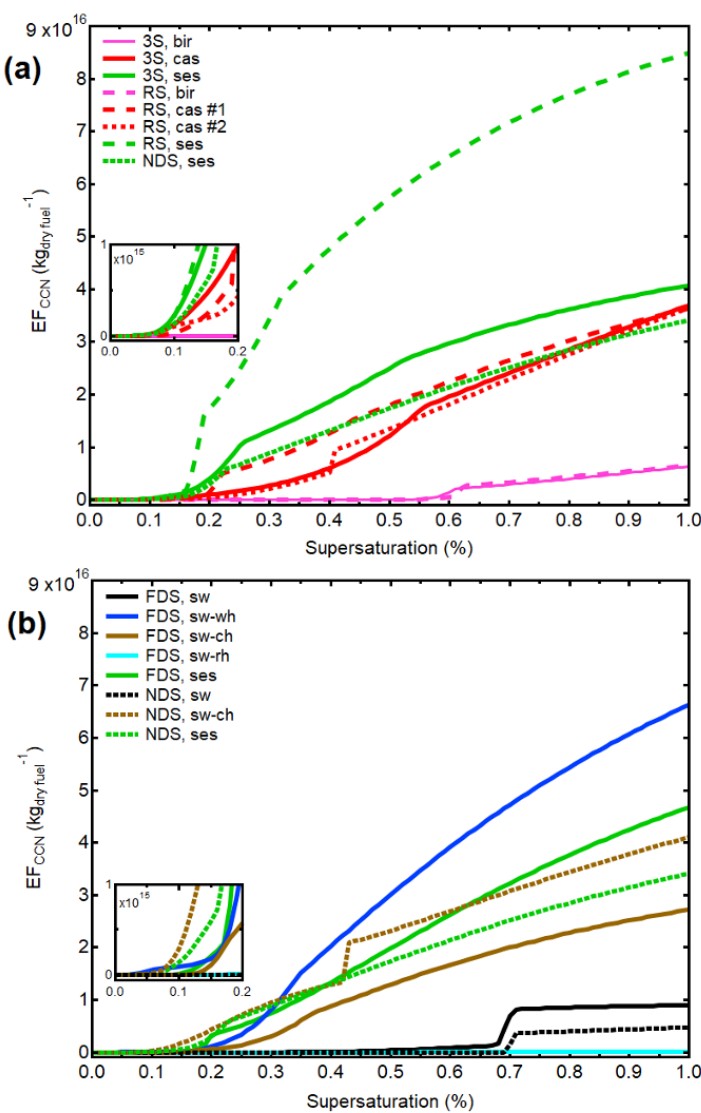

**Figure 4.** Normalised average 'primary' CCN emission factors vs supersaturation for various combinations of stoves and fuels. The curves are inferred from the average particle number size distributions during which the chamber was filled (Fig. 1) combined with the inter- and extrapolated CCN activity, which typically was measured for 3 (2-4) particle mobility sizes (Fig. 2). The NDS-ses emission factors are included in both (a) and (b) for comparison.



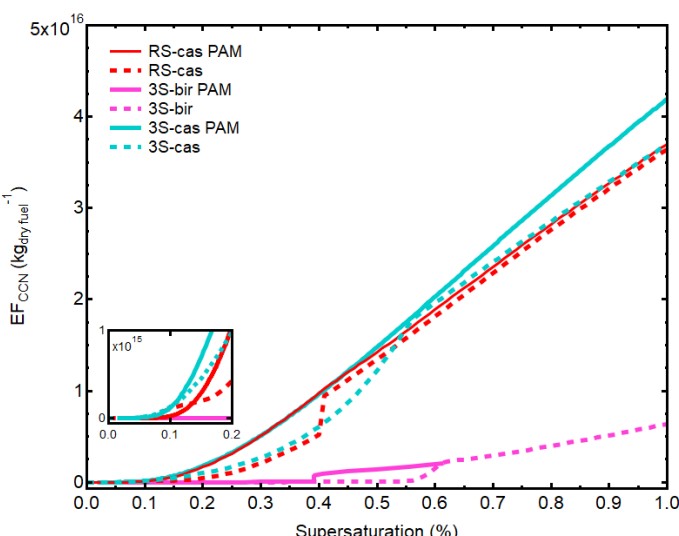

**Figure 5.** Examples of emission factors of CCN as estimated for the aerosol aged inside PAM with the unaged emission factors included for comparison. For the RS-cas PAM experiment, the nucleation particle mode grew large enough to dominate near a mobility diameter of 65 nm, which resulted in an increase in CCN for high supersaturations. For most other PAM experiments, it was not possible to estimate the isolated effect of ageing on the ultrafine particles with a reasonable certainty due to (i) the nucleation mode not growing large enough for CCN sizes, and (ii) coagulation during storage in the aerosol storage chamber.



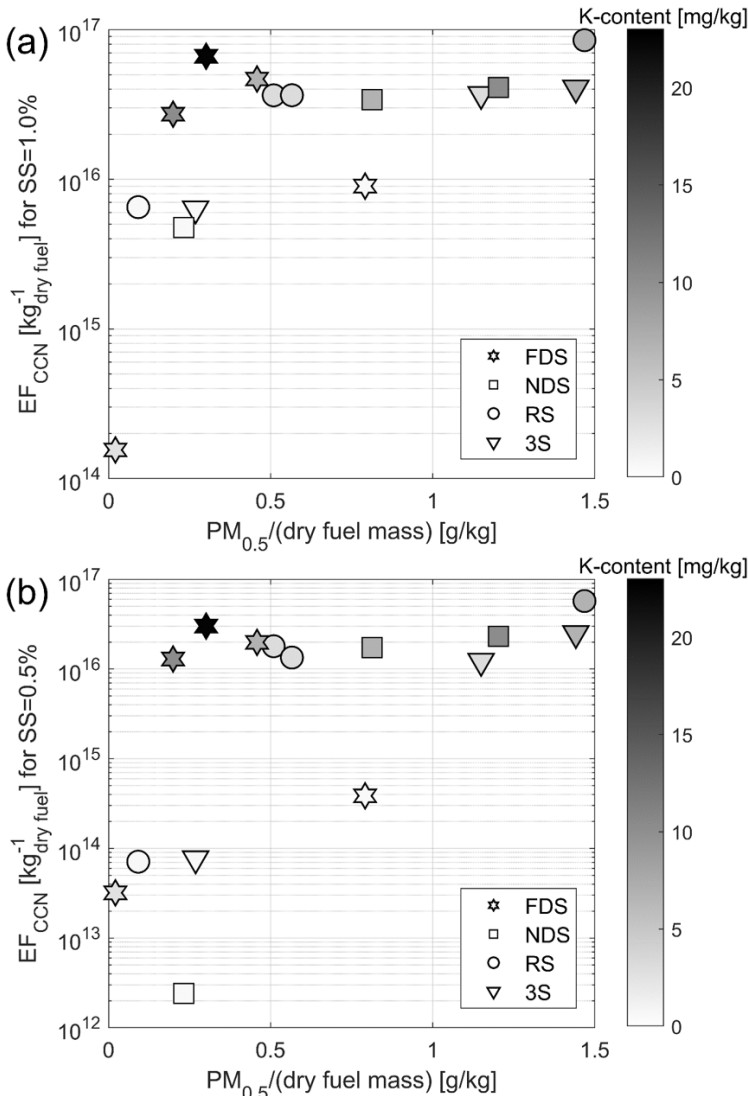

**Figure 6.** Primary emission factors (EFs) of cloud condensation nuclei (CCN) normalised to dry fuel consumption for a supersaturation (SS) of 1.0% in (a) and 0.5% in (b), respectively, versus estimated EFs of $PM_{0.5}$. The colourbar indicates the potassium concentration in the dry fuel. The CCN EFs are presented for a larger range of SSs in Fig. 4 and more details about the inorganic composition of the fuels can be found in Table 3. The EFs of $PM_{0.5}$ are estimated from the particle number size distributions shown in Fig. 1 in conjunction with effective density measurements as presented in Fig. 3. It should be noted that the range of the ordinate differs between the two subfigures.



**Table 1.** The experiments involving CCN measurements. For the 3-stone (3S), the rocket stove (RS), the natural draft stove (NDS) and the forced draft stove (FDS) in combinations with wood logs of birch (bir), casuarina (cas) and sesbania (ses), and pellets of softwood (sw), coffee husk (ch), rice husk (rh) and water hyacinth (wh). The numbers in the table refer to the number of experiments carried out for the respective combination of stoves and fuels.

| Stove\fuel | bir | cas | ses | sw | $ch^a$ | $rh^a$ | $wh^a$ |
|---|---|---|---|---|---|---|---|
| 3S | 1 | $3^b$ | 1 | - | - | - | - |
| RS | 1 | $5^b$ | 1 | - | - | - | - |
| NDS | - | - | $1^c$ | $2^b$ | 1 | - | - |
| FDS | - | - | $1^c$ | 3 | $2^b$ | 1 | 1 |

[a] In 50%-50% mixtures with sw by mass. [b] Including sampling through PAM. [c] Pelletized.

**Table 2.** The measured CCN activity ($\kappa$) of freshly emitted aerosol particles. The reported ranges represent $\pm 2$ standard deviations for the variation between independent measurements.

| Exp\$D_p$ [nm] | ⌣20-30 | 65 | 100 | 200 |
|---|---|---|---|---|
| 3S-bir | N/A | 0.07±0.01 | 0.04±0.00 | 0.006±0.002 |
| 3S-cas | ⌣0.17 | 0.14±0.05 | 0.07±0.03 | 0.05±0.02 |
| 3S-ses | N/A | 0.25±0.03 | 0.23±0.01 | 0.08±0.01 |
| RS-bir | N/A | 0.18±0.01 | 0.04±0.01 | 0.006±0.001 |
| RS-cas#1 | ⌣0.40 | 0.36±0.07 | 0.31±0.03 | 0.05±0.00 |
| RS-cas#2 | ⌣0.40 | 0.35±0.06 | 0.06±0.02 | 0.01±0.00 |
| RS-ses | N/A | 0.68±0.10 | 0.42±0.06 | 0.07±0.01 |
| NDS-ses | N/A | N/A | 0.29±0.01 | 0.07±0.04 |
| NDS-sw | ⌣0.12 | 0.09±0.04 | N/A | 0.001±0.000 |
| NDS-sw-ch | N/A | 0.38±0.13 | 0.07±0.02 | 0.013±0.003 |
| FDS-ses | N/A | N/A | 0.37±0.03 | 0.06±0.02 |
| FDS-sw | ⌣0.20 | 0.04±0.01 | 0.03±0.01 | 0.004±0.002 |
| FDS-sw-ch | N/A | 0.51±0.17 | 0.28±0.0 | 0.12±0.01 |
| FDS-sw-rh | ⌣0.12 | 0.11±0.01 | 0.07±0.01 | 0.02±0.01 |
| FDS-sw-wh | N/A | 0.58±0.14 | 0.39±0.04 | 0.20±0.02 |





**Table 3.** Ash content and major ash forming elements on a dry fuel basis.

| Species\fuel | bir | cas | ses | sw | ch | rh | wh |
|---|---|---|---|---|---|---|---|
| Ash [wt%] | 0.42 | 2.3 | 2.4 | 0.3 | 5.2 | 26.4 | 25.5 |
| Al [mg/kg] | 5 | 40 | 121 | 68 | 298 | 1460 | 13000 |
| Ca [mg/kg] | 1050 | 6350 | 3550 | 923 | 4080 | 1150 | 16900 |
| Cl [mg/kg] | 17 | 2.3 | 2.4 | 34 | 535 | 1060 | 33200 |
| Fe [mg/kg] | N/A | 55 | 121 | 43 | 312 | 1990 | 8870 |
| K [mg/kg] | 600 | 3100 | 6950 | 455 | 20200 | 4920 | 44800 |
| Mg [mg/kg] | 202 | 580 | 510 | 141 | 1090 | 606 | 6630 |
| Na [mg/kg] | 2.6 | 162 | 53 | 31 | 56 | 134 | 3040 |
| P [mg/kg] | 143 | 480 | 154 | 56 | 1000 | 944 | 3730 |
| S [mg/kg] | 118 | 2.3 | 2.4 | 57 | 1190 | 448 | 3290 |
| Si [mg/kg] | <25 | 117 | 555 | 189 | 657 | 109000 | 31000 |
| Ti [mg/kg] | N/A | 4 | 9 | 2 | 45 | 2640 | 717 |
| Zn [mg/kg] | 45 | 3 | 7 | 10 | 19 | 19 | 38 |





**Table 4.** Some hygroscopicity parameter ($\kappa$) values for selected species potentially relevant for the biomass combustion aerosol emissions studied.

| Species | $\kappa_{CCN}$ | $\kappa_{GF}$ | reference |
|---|---|---|---|
| HCl | $1.1^a$ | | Kristensen et al. (2012) |
| CaCl$_2$ | 0.48 | | Sullivan et al. (2009) |
| Ca(NO$_3$)$_2$ | 0.51 | | Sullivan et al. (2009) |
| CaSO$_4$ | 0.01 | | Sullivan et al. (2009) |
| CaCO$_3$ | 0.01 | | Sullivan et al. (2009) |
| CaO | ⌣0.00 | | Carrico et al. (2010) |
| KCl | | 0.99 | Carrico et al. (2010) |
| K$_2$CO$_3$ | | 0.7 | Rissler et al. (2005) |
| KNO$_3$ | | 0.93 | Carrico et al. (2010) |
| K$_2$SO$_4$ | 0.55 | 0.52 | Slade et al. (2015) |
| H$_2$SO$_4$ | 0.7 | | Shantz et al. (2008) |
| SiO$_2$ | <0.01 | | Kumar et al. (2009) |
| POA | $0.09^b$ | | Engelhart et al. (2012) |
| SOA | $0.10^b$ | | Engelhart et al. (2012) |
| SOA | 0.10 | | this study |
| soot | $<0.01^c$ | | Wittbom et al. (2014) |

[a] Modeled.

[b] Average $\kappa$ for primary organic aerosol (POA) (estimate) and secondary organic aerosol (SOA) as defined in the experimental approach applied by Engelhart et al. (2012).

[c] Freshly emitted, uncoated and insoluble soot particles.





**Table 5.** CCN activity ($\kappa$) and effective density ($\rho_{eff}$) of ⌣65 nm secondary particles, and ⌣200 nm primary and aged particles, respectively. Inferred $\kappa$ values for secondary organic aerosol dominated particles are included for the three cases observed. The indicated ranges are ±2 standard deviations for the 200 nm primary particles when more than 2 data points are available, while they represent the full range observed for the more aged particle parameters.

| $D_p$ [nm] | 65 | | 200 | | | |
| --- | --- | --- | --- | --- | --- | --- |
| Aerosol | SOA | | primary | | aged | |
| Property | $\kappa$ | $\rho_{eff}$ | $\kappa$ | $\rho_{eff}$ | $\kappa$ | $\rho_{eff}$ |
| Exp\Unit | | [gcm$^{-3}$] | | [gcm$^{-3}$] | | [gcm$^{-3}$] |
| 3S-bir | N/A | 1.43±0.01 [b] | 0.006 | 0.71±0.01 | 0.012 | 0.82 |
| 3S-cas#1 | 0.11-0.14[a] | 1.36±0.01[a] | 0.050±0.002 | 0.87±0.07 | 0.06 | 0.96±0.03 |
| 3S-cas#3 | 0.12[a] | 1.40±0.01[a] | 0.041±0.012 | 0.85±0.01 | 0.09±0.02 | 0.92 |
| RS-cas#2 | N/A | N/A | 0.013±0.005 | 0.41±0.01 | 0.037±0.004 | 0.50±0.01 |
| RS-cas#3 | N/A | 1.40[b] | 0.070±0.022 | 0.45±0.01 | 0.077±0.007 | 0.64±0.03 |
| RS-cas#4 | N/A | 1.40[b] | 0.033 | 0.47±0.02 | 0.055±0.007 | 0.54±0.01 |
| RS-cas#5 | N/A | 1.41±0.01[a] | 0.071 | N/A | 0.094±0.007 | 0.87±0.01 |
| NDS-sw | 0.10-0.12[a] | 1.27[a] | 0.001 | 0.50±0.01 | N/A | 0.65±0.01 |
| FDS-sw-ch | N/A | N/A | 0.12±0.01 | 0.41±0.02 | 0.14±0.03 | 0.53±0.02 |

[a] Dominated by secondary aerosol.

[b] Particles comprised of a mixture of primary and secondary aerosol.