# Peer review of "Properties and emission factors of CCN from biomass cookstoves -observations of a strong dependency on potassium content in the fuel"

_Atmospheric Chemistry and Physics, 2020_

## Referee Comment (RC1) · Anonymous Referee #1 · 7 Dec 2020

The manuscript discusses the properties of cloud condensation nuclei (CCN) emitted from biomass burning of solid fuels (seven different fuels) in different cookstoves (four different stoves). This study covers particle number size distribution, mixing state, particle density, chemical composition, CCN activation and particle hygroscopicity properties at the same time. The measurement results offer valuable insights for field measurements and global models regarding biomass burning particles.

Overall, the paper is well written and relevant to ACP. I recommend publication after the following comments are satisfactorily addressed:

Major comments:

1. There were a bunch of measurements clearly explained in the manuscript. But I think it would be nice to have an overview plot or measurement setup sketch. It should include biomass burning setup, chambers, sampling line, and instruments. It will help readers to understand your measurements better. A simple example can be found in Smith *et al.*, (2019).

2. Did you consider the particle wall losses and particle loss inside the inlets (diffusion, deposition, etc.)? For example, the emitted aerosols were injected into a chamber for 10-40 minutes. What is the wall loss affection of the size distribution?

Minor comments:

Line 205: I could not get why "aerosol particles present in the flue gas and initially injected into the aerosol storage chamber as freshly formed or primary, while particulate matter formed in the flow reactor will be considered secondary aerosol.". Could you please give more explanation about your definition of freshly formed and secondary aerosol?

Lines 241-242: Why the soot mode particle was unaffected in the storage for up to 60 minutes? Could you find previous studies that also support this? At least from your measurement, the aged

200 nm particles always had different (most probably higher) kappa values than primary particles.

Lines 269-270: It is good to adjust the PNSD for dilution rates and normalized to the corresponding consumption of dry fuel mass. I would suggest including the error bars in Fig. 1. The error bars could be 25$^{th}$ and 75$^{th}$ percentiles (with median lines) or one stand deviation (with mean lines). This will help us to understand the fluctuation of PNSD during the experiments.

Lines 305-308: Did you see relative higher slopes of the CCN activation spectra for the FDS, sw and NDS, sw-ch? Clear ultrafine and soot modes were observed for these two types of biomass burning particles. There is an overlap of ultrafine and soot modes in the size range around 100 nm. If we assume the ultrafine and soot mode particles have different chemical compositions, a relatively higher slope would be expected.

References:

Smith, D. M. *et al.* (2019) 'Construction and Characterization of an Indoor Smog Chamber for Measuring the Optical and Physicochemical Properties of Aging Biomass Burning Aerosols', *Aerosol and Air Quality Research*, 19(3), pp. 467–483. doi: 10.4209/aaqr.2018.06.0243.

---

## Referee Comment (RC2) · Anonymous Referee #2 · 11 Jan 2021

Review of 'Properties and emission factors of CCN from biomass cookstoves - observations of a strong dependency on potassium content in the fuel', T. Kristensen et al., ACPD, 2020

This manuscript presents an analysis of the CCN properties of the combustion of a range of fuels in different types of stoves to assess the impact of residential cooking fires on CCN emissions. This manuscript of part the Salutary Umea STudy of Aerosols IN biomass cookstove Emissions (SUSTAINE) to study sustainable approaches to residential cooking in Sub-Saharan East Africa. The experiments were conducted using

traditional stoves (3-stone stove and rocket stove) with wood logs comprised of sesbania, casuarina and birch (reference-type). Natural draft and forced draft stoves were used in conjunction with pellet fuel made with different materials, including coffee husk, rice husk or water hyacinth mixed with Swedish softwood, which also served as a reference. The objective of the experiment is to determine the contribution of stove and fuel to CCN emissions and assess the contribution of residential cooking to aerosol-cloud interactions and public health.

Measured particle number size distributions were bi-model with an ultrafine mode that was often less than 65 nm diameter, as well as a soot mode centered on average 150 nm diameter. CCN hygroscopicity and effective density were measured for up to four sizes (65, 100, 200, 350 um diameter). The ultrafine mode was moderate to highly hygroscopic (kappa between 0.1 and 0.8), with higher kappa associated with higher combustion temperature and soluble salts such as potassium, while nthe soot mode was much less hygroscopic (kappa between 0 and 0.15) and comprised of black carbon and organic material.

The correlation between CCN emission factors and PM emissions factors is highly dependent on aerosol hygroscopicity (particularly K content) and combustion temperatures (stove type), respectively. In general, well written, but at some points that need clarification, particularly the calculation of CCN emissions factors.

General comments: The reviewer suggests the authors add a schematic showing the layout of the instrumentation and the sampling configuration. A schematic would help distinguish which instruments are sampling in the flue, which instruments are sampling from the stainless-steel chamber, and how the dilution lines and ageing experiments are implemented. In addition, details such as lengths and layout of sampling lines and chamber residence times are helpful in assessing particle losses. Pertinent details should be included in this manuscript and not just referred to in another publication.

In lines 240+, the authors state that they were not able measure an increase in diameter with ageing from particle number size distribution in spite of such indications from the CCN, APM and AMS measurements. Yet, in the same section, the authors note that that average aerosol number size distributions were larger than flue measurements. Much of the discussion in this manuscript on ageing was centered around chemical changes by the addition of secondary organic material, but there are also physical changes to soot particles, particularly in the first hours after emissions [Li et al., Atmospheric Environ., 2015; doi: 10.1016/j.atmosenv.2015.09.003]. This section would benefit from a discussion of the evolution of particle morphology as well.

In lines 519+, the authors state that the step in the FDS-sw curve near 0.7% supersaturation (Figure 4b) is related to activation of the soot mode within a narrow supersaturation range. The FDS-sw soot mode (Figure 1) appears encompass a size range from ca. 80 to 250 nm diameter, which corresponds to supersaturations between approximately 1.6% to 0.4%, respectively (kappa $\sim$ 0.004). This range of supersaturations for the soot mode is more than an order of magnitude more than the range shown in the step in Figure 4b. The reviewer suspects there is a step in the CCN spectra related to a discontinuity between flow scans and CCN spectra or the use of discrete hygroscopicity values applied to the number size distribution. The authors need to describe their calculations.

Specific comments: L11: specify or give a reference for 'standard protocols'

L43+: it would help the read to specify 'diameter' throughout the text

L77: change 'focus' to its plural 'foci'

L115: triangular cross-section has been described with two numbers (dimensions of 2.6 cm times 2.5 cm . . .). It's not clear to what these dimensions are referring.

L124: provide a reference for the standardized water boiling test 4.2.4

L130: replace 'were operated' with 'were initiated' or 'began'

L170: what was the range of the total flow rate scans of the CCN instrument?

L276+: The authors state that particle number size distribution in the flue (from the fast particle analyser) were similar to the initial measurements taken by the SMPS in the aerosol chamber, which seems to confirm consistency between independent measurements of particle number size distributions. The authors then note than average number particle size distributions in the chamber were larger than those of the flue, which seems logical given that the particles in the chamber are no longer representative of the fresh emissions in the flue. Why do the authors speculate losses, coagulation in the sample lines or offset between instruments, especially given the consistency between the fast particle analyser and the SMPS at the beginning of the experiment?

L302, Section 4.2: The impact of mixing state can also be assessed by using the aerosol size distribution and associated aerosol hygroscopicity to regenerate the CCN spectra and compare cases of different mixing states. The authors inferred a single kappa value, which suggests then that either the aerosol is internally mixed or that external mixtures do not produce significant differences in the CCN spectra in this study. Also, what do the authors mean by CCN spectra of 'appropriate quality'? The reviewer also encourages the authors to add a figure showing the CCN spectra to compliment Figure 1.

L371: Do the authors mean a more spherical aggregate when referring to more compact black carbon particles? The ageing experiments using the PAM and thermodenuder clearly show the impact of the SOA condensing onto fractal aggregates. However, did the authors also observe morphological changes (for example, an evolution of effective density compared to initial measurements)?

L390; Section 4.3.2: A reference to Table 3 needs to be added earlier in the paragraph to orient the reader.

L397: replace 'totally' with a quantitative assessment

L403: replace 'state-of-the-art' with literature references

L407: remove 'more' in 'more refractory species'

L455+: This discussion in this section could be reorganized and main points clearly stated. This paragraph discusses 3S-RS results, the next paragraph discusses NDS and FDS results, and then the discussion returns back to 3S-RS results.

L473: What do the authors mean by 'variations in kappa for 200 nm particles . . .. correlate with variations in ultrafine kappa values for FDS'? The discussion in this paragraph is not clear. The reviewer interprets the results as 3S associated with a higher organic fraction across the entire size distribution and is consistent with higher relative densities, while RS have higher EC fractions along with lower relative densities.

L500: what 'special conditions' are the authors referring to?

L524: Have the authors tried to quantify wall losses and coagulation to assess how much they may impact CCN emissions?

L614: The authors suggest that PM emissions are sensitive to the very large particles, which is not entirely correct. PM emissions are sensitive to the mass size distribution (the product of the number concentration and the particulate mass at a given size). A figure showing the calculated mass size distribution using the effective densities would be useful in illustrating this point.

L664+: The reviewer suggests to integrate the perspectives (wildfires and health impacts) into the conclusions.

L676+: As stated in the text, the optimal scenario would be a reduction in both PM and CCN emission factors. Based on the experiments conducted here, can the authors reiterate what specific combinations of stove / fuel should and should not be used?

Figures 1 and 2: A description of the legend is needed in the figure captions.

Figures 4 and 5: As mentioned previously, the issue with the steps in the emission factors needs to be resolved. It is also not clear what is the purpose of the insets at

low supersaturation.

Figure 5: 'For most other PAM experiments...' This analysis needs to be in the main text rather than the figure caption.

Figure 6b: The upper part of the label for gray-scale bar has been cut.

Table 3: Add the chemical analysis used to determine the ash content in the figure caption.

---

## Author Comment (AC1) · 5 Mar 2021

We thank the anonymous reviewer for the many detailed comments and suggestions. We find that the revised version of the manuscript has improved due to these comments, and we thank the anonymous reviewers in the revised version of the manuscript. Our references to line numbers and figures generally refer to the ACPD version of the manuscript – unless specified differently. There appears to be an offset in line numbers of about 1 between the manuscript version the reviewer refers to, and the ACPD version of the manuscript, which we refer to. Our reponses appear in **bold**

[Figure]

below.

This manuscript presents an analysis of the CCN properties of the combustion of a range of fuels in different types of stoves to assess the impact of residential cooking fires on CCN emissions. This manuscript of part the Salutary Umea STudy of Aerosols IN biomass cookstove Emissions (SUSTAINE) to study sustainable approaches to residential cooking in Sub-Saharan East Africa. The experiments were conducted using traditional stoves (3-stone stove and rocket stove) with wood logs comprised of sesbania, casuarina and birch (reference-type). Natural draft and forced draft stoves were used in conjunction with pellet fuel made with different materials, including coffee husk, rice husk or water hyacinth mixed with Swedish softwood, which also served as a reference. The objective of the experiment is to determine the contribution of stove and fuel to CCN emissions and assess the contribution of residential cooking to aerosol-cloud interactions and public health. Measured particle number size distributions were bi-modal with an ultrafine mode that was often less than 65 nm diameter, as well as a soot mode centered on average 150 nm diameter. CCN hygroscopicity and effective density were measured for up to four sizes (65, 100, 200, 350 nm diameter). The ultrafine mode was moderate to highly hygroscopic (kappa between 0.1 and 0.8), with higher kappa associated with higher combustion temperature and soluble salts such as potassium, while the soot mode was much less hygroscopic (kappa between 0 and 0.15) and comprised of black carbon and organic material. The correlation between CCN emission factors and PM emissions factors is highly dependent on aerosol hygroscopicity (particularly K content) and combustion temperatures (stove type), respectively. In general, well written, but at some points that need clarification, particularly the calculation of CCN emissions factors.

General comments:
The reviewer suggests the authors add a schematic showing the layout of the instru-

mentation and the sampling configuration. A schematic would help distinguish which in-struments are sampling in the flue, which instruments are sampling from the stainless-steel chamber, and how the dilution lines and ageing experiments are implemented. In addition, details such as lengths and layout of sampling lines and chamber residence times are helpful in assessing particle losses. Pertinent details should be included in this manuscript and not just referred to in another publication.

**A schematic of the experimental setup has been included as the first figure in the revised manuscript.**

In lines 240+, the authors state that they were not able measure an increase in di-ameter with ageing from particle number size distribution in spite of such indications from the CCN, APM and AMS measurements. Yet, in the same section, the authors note that that average aerosol number size distributions were larger than flue mea-surements. Much of the discussion in this manuscript on ageing was centered around chemical changes by the addition of secondary organic material, but there are also physical changes to soot particles, particularly in the first hours after emissions [Li et al., Atmospheric Environ., 2015; doi: 10.1016/j.atmosenv.2015.09.003]. This section would benefit from a discussion of the evolution of particle morphology as well.

**We made lognormal mode fits to the SMPS soot mode before, during and after sampling through the oxidation flow reactor. In a few cases, we observed very minor increases in the soot-mode-diameter during parts of the simulated age-ing experiments, and in most cases we did not. The experiments included in the 'aged-CCN-emission factors-figure' mainly belong to the latter category. The fact that the CCN and effective density measurements were more sensitive to minor SOA formation on the soot particles than the SMPS measurements is not surprising. The initial kappa and effective density levels were significantly below the expected corresponding values for SOA, so addition of a relatively small SOA volume can be detected in those properties.**

**We have added the following comment to L247: "This is most likely due to the initial soot particle CCN activity and effective density being very low compared to those properties of SOA. Hence, soot particle CCN activity and effective density are very sensitive to minor additions of SOA."**

**The ageing effects reported by Li et al. (2015) are all based on a rather different aerosol stored in an environment with significantly higher RH than in our study. Furthermore, we generally report results obtained shortly after chamber filling. We cannot rule out effects of coagulation, which is discussed in detail in the paper – but as mentioned, that will be the case in any type of experimental set up – and also real-life cookstove emissions.**

**We mainly include the effects of photochemical ageing on the CCN properties on a qualitative level, and we do not find it adequate to go further into detail with such ageing effects in this study. The experimental approach was simply not optimized for studies of photochemical ageing.**

In lines 519+, the authors state that the step in the FDS-sw curve near 0.7% supersaturation (Figure 4b) is related to activation of the soot mode within a narrow supersaturation range. The FDS-sw soot mode (Figure 1) appears encompass a size range from ca. 80 to 250 nm diameter, which corresponds to supersaturations between approximately 1.6% to 0.4%, respectively (kappa=0.004). This range of supersaturations for the soot mode is more than an order of magnitude more than the range shown in the step in Figure 4b. The reviewer suspects there is a step in the CCN spectra related to a discontinuity between flow scans and CCN spectra or the use of discrete hygroscopicity values applied to the number size distribution. The authors need to describe their calculations.

**The calculations behind the curve was described in L. 220-224, and the relevant kappa values for 100 nm and 200 nm particles were provided in Table 2. The reviewer assumes the kappa value for the soot mode to be constant with chang-**

ing mobility diameters, which is not a reasonable assumption. Nevertheless, the reviewer is both right and wrong here.

It is not meaningful to assume a constant apparent kappa value of 0.004 for the entire soot mode. Nevertheless, this reviewer comment led us to reprocess a lot of data, and we found a mistake in the data processing influencing this particular CCN emission factor curve. We will describe these aspects in more detail below.

In this study, we report kappa-values calculated from the mobility diameter as described. There are 3 reasons why we would expect such a kappa value to decrease with increasing mobility size over a soot mode, which is evident from the kappa values reported in Table 2.

1. If we assume that the chemical composition of the soot mode does not vary significantly with the particle size, then one may assume that the kappa-value calculated from the volume equivalent diameter could be close to constant (Dusek et al., 2011). We observe pronounced decreases in the effective density with increasing soot particle mobility diameter (Fig. 3). Hence, for that reason, the reported kappa values should decrease with mobility size over the soot mode. However, that shape-effect is unlikely to explain the entire difference in reported kappa values between the 100 nm and the 200 nm soot particles in our study.

2. Kappa is a volume-based property as mentioned in the manuscript. We observe strong indications of the ultrafine particles contributing to increasing the kappa-values of the soot modes. The kappa value of the 100 nm soot particles will increase 'faster' with coagulation with the ultrafine particles relative to the much larger volume of the 200 nm soot particles. We expect this effect to be of importance in this study, but it is not straightforward to estimate the magnitude of this effect.

3. There is a third effect leading to expected decreasing kappa with mobility diameter for a soot mode, which is described in more detail by Wittbom et al.

(2014). For low-hygroscopicity soot particles, the cloud droplet activation may occur on just a fraction of a soot particle since the critical droplet size at activation is relatively small. That effect will be more pronounced with increasing soot particle size and lower kappa value, and it depends on particle morphology. Whether this effect plays a role for the CCN activity when comparing kappa values of 100 and 200 nm soot particles in our study is not entirely clear, but based on the work by Wittbom et al. (2014), it cannot be ruled out.

In the re-processing of our data, we discovered a mislabeling of a data file. Unfortunately, the reported kappa values for the FDS-sw represented an experiment with an alternative and non-ideal stove operation. Hence, that mistake influenced the CCN emission factor curve of the FDS-sw experiment presented in Fig. 4.b. The correct values for the FDS-sw experiment are now presented in Fig. 2, Fig. 4.b and Table 2 of the manuscript.

In this case, the reviewer was correct that the pronounced step in the CCN emission factor curve previously reported – should indeed not be as steep. But it does not change the fact that significant mobility size ranges of the soot mode often activate within relatively narrow supersaturation intervals due to the reasons described above.

We have replaced:
"The step in the FDS-sw curve near SS=0.7% is due to activation of the entire soot mode within a narrow range of supersaturations. That soot mode is probably produced due to quenching of the flame on the bottom of the pot, and a more optimal fuel load can potentially significantly reduce those soot particle emissions."
With:
"The steps observed in some cases (e.g. NDS-sw, NDS-sw-ch) are results of significant particle size ranges activating into cloud droplets over a narrow range of supersaturations due to the strong increase in kappa with decreasing Dp in

**some cases."**

Specific comments:

L11: specify or give a reference for 'standard protocols'

**We do not find it adequate to provide references to standard protocols in the abstract. Specifications are provided in the relevant section (Materials and Methods). The analysis of the fuel composition was carried out in a specialized and commercial external laboratory. The protocols can be purchased online.**

L43+: it would help the read to specify 'diameter' throughout the text

**The mobility diameter ($D_p$) is defined in L. 88 and used throughout the manuscript. The diameter referred to in L. 43 would typically be the aerodynamic diameter, and that is not of direct relevance to the focus of our study.**

L77: change 'focus' to its plural 'foci'

**That has been changed as suggested**

L115: triangular cross-section has been described with two numbers (dimensions of 2.6 cm times 2.5 cm . . .). It's not clear to what these dimensions are referring.

**Those dimensions refer to the base and the height, respectively. The text has been changed to include that information.**

L124: provide a reference for the standardized water boiling test 4.2.4

**We find it more appropriate to refer to the 4.2.3 water boiling test described by the Clean Cooking Alliance in 2014. The document can easily be found by use of a web search engine: https://www.cleancookingalliance.org/binary-data/DOCUMENT/file/000/000/399-1.pdf
But we do not know whether the document will remain there for a foreseeable future, so in the revised manuscript we provide the main web-address of the**

**organization.**

L130: replace 'were operated' with 'were initiated' or 'began'

**'were operated' has been changed to 'were initially loaded'**

L170: what was the range of the total flow rate scans of the CCN instrument?

**The range in CCNC flow rates (0.2 to 1.0 lpm) was described in detail in L. 155-160. The associated range in terms of supersaturation was between about 0.1% to about 1.5% depending on the temperature gradient as reported in L. 169-170.**

L276+: The authors state that particle number size distribution in the flue (from the fast particle analyser) were similar to the initial measurements taken by the SMPS in the aerosol chamber, which seems to confirm consistency between independent measurements of particle number size distributions. The authors then note than average number particle size distributions in the chamber were larger than those of the flue, which seems logical given that the particles in the chamber are no longer representative of the fresh emissions in the flue. Why do the authors speculate losses, coagulation in the sample lines or offset between instruments, especially given the consistency between the fast particle analyser and the SMPS at the beginning of the experiment?

**The two different instruments (FPA and SMPS) used for measuring particle number size distributions did not produce 1:1 identical particle number size distributions (as described in more detail in the text), and we would not expect them to do so. The general particle number size distribution features were rather similar between the two instruments – when comparing the flue gas measurements with the initial emissions into the aerosol storage chamber – as stated in the text. It was not possible for us to tell to which extent the described minor differences were due to instrumental differences and/or diffusional losses/coagulation effects.**

**The aerosol population in the flue gas orgininating from biomass combustion**

**may change significantly over timescales at the order of 10s, so the FPA is the better choice for measurements in the flue gas. In addition, the instrument is optimised to reduce losses of the smallest particles. We have included the following to section 3.3 in order to clarify advantages and limitations to the FPA in this context.**

**"It was operated with a sample flow rate of 8.0 slpm and the inversion matrix for bimodal spherical aerosol particles from the manufacturer was applied. This matrix is suitable for spherical biomass particles and heavily coated soot, but it may underestimate the sizes of fractal-like soot particles due to the particle shape influencing the charging efficiency. That effect may play a role for fractal soot aggregates with mobility diameters larger than 150 nm and can, when these are present, result in a slight overestimation of particle numbers at smaller sizes (Symonds et al., 2007). The main advantages of the FPA when studying combustion aerosol emissions online are (i) a very fast instrumental response time (<1s), and (ii) minimised losses of the smallest particles (Reavell et al., 2002)"**

**The main point is that minor biases in the particle number size distributions are unlikely to substantially impact any of the reported CCN results. Thus we have added the following statement in the L276+: "It is worth noting that none of the listed potential minor effects will have substantial influence on the results reported in this study."**

L302, Section 4.2: The impact of mixing state can also be assessed by using the aerosol size distribution and associated aerosol hygroscopicity to regenerate the CCN spectra and compare cases of different mixing states. The authors inferred a single kappa value, which suggests then that either the aerosol is internally mixed or that external mixtures do not produce significant differences in the CCN spectra in this study. Also, what do the authors mean by CCN spectra of 'appropriate quality'? The reviewer also encourages the authors to add a figure showing the CCN spectra to compliment Figure 1.

Yes, in principle, we agree with the reviewer regarding links between CCN spectra and mixing state as described in L. 305-309. However, in this context we find it essential to draw the attention to L. 310-311: 'However, it should also be mentioned that the size resolution obtained with the DMA in the current study was not optimised for identification of the CCN mixing state' The CCNC experimental approach was optimized for obtaining CCN spectra very fast in parallel with the APM measurements. A different approach should had been applied, if our focus had been on extracting detailed information from CCN spectra. We have rephrased the statement in L.310-311 to: 'However, it should be noted that the CCNc experimental approach was optimised for fast scans in parallel with the APM measurements. Hence, neither the DMA transfer function nor the CCNc operation were optimised for extracting information about the mixing state from the CCN spectra.'

Furthermore, it is also worth noting that the reported main findings on CCN activity and associated indications of chemical composition of particle modes, as well as CCN emission factors may be rather sensitive to measurements at 65 and 200 nm, while it is often less sensitive measurements at 100 nm. None of the main findings in our study are sensitive to the reported kappa values for 100 nm particles.

In summary, neither do we find it justifiable to present fast scan CCN spectra in this paper as results in a separate figure, nor do we find such an approach essential for the reported main results.

When it comes to CCN spectra of 'appropriate quality', we have added the following to section 3.4:
'CCN spectra were excluded from the data analysis for four different reasons: (i) if the dT was not constant, (ii) if a full CCN spectrum was not covered within the scan range, (iii) if the CCN counting statistics and signal to noise ratio were too low for analysis, or (iv) in rare cases, if the actual CCNC flow profile over a scan

differed significantly from the pre-programmed profile.'

**As a consequence, L. 308-310: have been shortened from:**
**'Regardless of which of the listed options that can explain our observations, we were generally able to infer a single CCN activity from the CCN spectra of appropriate quality in terms of reasonable and stable dT and reasonable CCN counting statistics.'**

**To:**
**'Regardless of which of the listed options that can explain our observations, we were generally able to infer a single CCN activity from the analysed CCN spectra.'**

L371: Do the authors mean a more spherical aggregate when referring to more compact black carbon particles? The ageing experiments using the PAM and thermodenuder clearly show the impact of the SOA condensing onto fractal aggregates. However, did the authors also observe morphological changes (for example, an evolution of effective density compared to initial measurements)?

**We have rephrased 'more compact black carbon particles' to 'more compact black carbon aggregates'. In most cases, 'more compact' is likely to imply 'more spherical' – but shapes may be far from spherical. 'Compact' is widely used in context with such soot properties in the literature.**

L390; Section 4.3.2: A reference to Table 3 needs to be added earlier in the paragraph to orient the reader.

**The following statement has been added to L390:**
**"The concentration of the dominant inorganic species in the applied fuels are presented in Table 3"**

L397: replace 'totally' with a quantitative assessment

**The sentence has been rephrased to: "For rh, the fuel ash was dominated by Si (>85% by mass) with a minor fraction of K ( 4% by mass)."**

[Figure]

L403: replace 'state-of-the-art' with literature references

**'state-of-the-art' has been deleted, and references have been added at the end of the sentence (Obernberger et al., 2006; Bostrom et al., 2011; Obaidullah et al., 2012.)**

L407: remove 'more' in 'more refractory species'

**'more' has been deleted as suggested.**

L455+: This discussion in this section could be reorganized and main points clearly stated. This paragraph discusses 3S-RS results, the next paragraph discusses NDS and FDS results, and then the discussion returns back to 3S-RS results.

**We agree, and we have restructured the section for improved clarity.**

L473: What do the authors mean by 'variations in kappa for 200 nm particles . . .. correlate with variations in ultrafine kappa values for FDS'? The discussion in this paragraph is not clear. The reviewer interprets the results as 3S associated with a higher organic fraction across the entire size distribution and is consistent with higher relative densities, while RS have higher EC fractions along with lower relative densities.

**The discussed correlation is of general relevance to all the studied stoves. We have rephrased to:**
**"Whenever kappa values for the 200 nm particles were relatively low (or high), they were associated with relatively low (or high) kappa values for the ultrafine mode ( 65 nm). That relation is represented by a correlation coefficient of 0.76 and a p-value of 0.01 for the results presented in Fig. 3. Hence, it is likely that the ultrafine particles coagulated with the soot mode particles and thus increased the soot mode kappa values. These observations further support, that not only do the potassium compounds appear to play a key role for the CCN activity of the ultrafine particles, they also influence the CCN activity of the soot mode."**

L500: what 'special conditions' are the authors referring to? For the RS-ses experiment, the chamber injections did not represent a full combustion cycle. These 'special conditions' were described in more detail in L. 295-303.

**'which to some extent is likely to be due to the special conditions for the chamber injections for the RS-ses experiment' has be rephrased to:**
**'which to some extent is likely to be due to the chamber injections for the RS-ses experiment not being representative of a full combustion cycle as described above.'**

L524: Have the authors tried to quantify wall losses and coagulation to assess how much they may impact CCN emissions?

**We did not quantify particle losses prior to measurements of the particle number size distributions in the flue gas, but we consider potential losses negligible when it comes to the presented CCN emission factors and $PM_{0.5}$ results. The particle number size distributions presented were measured in the flue gas close to the stoves. Flow rates were high in the flue gas (2.4 m3/min) and in the short sampling line for the fast particle analyser (8 slpm). Hence, we consider diffusional losses to be very minor. There may be non-negligible diffusional losses of the very smallest particles potentially influencing the 'left part' of the presented ultrafine particle modes for e.g. the FD-sw and the FD-sw-rh. However, such small particles will neither act as CCN for atmospherically relevant supersaturations nor contribute significantly to the $PM_{0.5}$. So we do not expect particle losses to influence any of the main results presented.**

**Coagulation will play a role in any kind of experimental set up to study CCN emissions from cookstoves – as well as in real-life settings. We find that our approach with significant dilution in the hood and sampling with the fast particle analyser after 2s residence time in the flue gas is optimal for assessment of the primary emissions with minimal impact of coagulation on the particle number size distributions. Coagulation involving the soot particles increases the kappa**

value of those particles, which is clearly stated in the manuscript. It is not possible for us to assess the importance of coagulation immediately above the flames versus potential coagulation in sampling lines. Hence, we discuss these effects on a qualitative level in more detail in the manuscript.

L614: The authors suggest that PM emissions are sensitive to the very large particles, which is not entirely correct. PM emissions are sensitive to the mass size distribution (the product of the number concentration and the particulate mass at a given size). A figure showing the calculated mass size distribution using the effective densities would be useful in illustrating this point.

**In the paragraph referred to, we discuss the contribution of the soot mode to the PM$_{0.5}$ emissions from the FDS-sw and the RS-ses. Based on integration of the estimated particle mass distributions, the particles with mobility diameters >100 nm contributed 97% and 74% to the PM$_{0.5}$ emissions for the FDS-sw and the RS-ses, respectively. Hence, the PM$_{0.5}$ emissions were indeed sensitive to reductions in the soot mode in those cases. We do not find reason to revise the paragraph. Furthermore, we do not find reason to include additional figures with particle mass size distributions, since it was not a focus area of this study. The inclusion of estimated PM$_{0.5}$ emissions was mainly motivated by illustrating the challenge of linking PM levels to CCN concentrations for the studied aerosol. We consider that of importance to the modeling community.**

L664+: The reviewer suggests to integrate the perspectives (wildfires and health impacts) into the conclusions.

**We have added the following to the conclusion:
"Overall, we observed high potential to significantly reduce primary PM$_{0.5}$ emissions from biomass cookstoves by (i) applying fuels with low levels of inorganic compounds potentially entering the aerosol phase, (ii) improved stove technology, and (iii) optimal stove operation. Reduced PM and soot emissions are mo-**

**tivated from health and climate perspectives. However, our study indicates that biomass fired cookstoves may comprise a very significant and underestimated source of CCN, and substantial reductions of such CCN emissions may potentially lead to warming effects on climate depending on the prevailing meteorological conditions and importance relative to other significant CCN sources. Our study indicates, that with the right combination of stove and fuel, it is possible to significantly reduce the soot and PM emissions while maintaining pronounced emissions of highly hygroscopic ultrafine particles - depending on the fuel. However, stove costs, infrastructures and fuel quality, availability and sustainability have to be considered in the overall guidelines towards improved cookstoves. The observed strong impact of potassium on CCN emissions for a wide range of combustion conditions is likely to be of relevance not only to cookstoves but a wider range of biomass combustion including wild fires."**

L676+: As stated in the text, the optimal scenario would be a reduction in both PM and CCN emission factors. Based on the experiments conducted here, can the authors reiterate what specific combinations of stove / fuel should and should not be used?

**We do not state in the text that 'the optimal scenario would be a reduction in both PM and CCN emission factors' – it may be more complicated than that. The soot mode dominated the $PM_{0.5}$, and reduced emissions of soot particles can be desired from both a health and a climate perspective. It is less clear to which extent the more hygroscopic ultrafine particles may pose a significant health issue, and it may very well be that such emissions dominate the CCN population in large regions. Hence, significantly reduced emissions of the ultrafine CCN could potentially lead to a pronounced net warming of climate – and that may not be a desired effect.**
**See our previous response above for the changes made to the conclusion.**

Figures 1 and 2: A description of the legend is needed in the figure captions.

[Figure]

The following has been added to the Fig. 1 caption:

'The abbreviations of stoves and fuels in the legends are: 3-stone fire (3S), rocket stove (RS), natural draft stove (NDS) and forced draft stove (FDS) with combustion of birch (bir), casuarina (cas) or sesbania (ses) wood logs or pellets of softwood (sw), coffee husk (ch), rice husk (rh), water hyacinth (wh) or ses.'

In the Fig. 2 caption, 'In (a), for the rocket stove (RS) and the 3-stone (3S) stove, in (b), for the forced draft pellet stove.'

**Has been rephrased to:**

**'In (a), for the rocket stove (RS) and the 3-stone (3S) stove with combustion of wood logs of birch (bir), casuarina (cas) or sesbania (ses). In (b), for the forced draft stove (FDS) and combustion of pellets of ses or softwood (sw) or mixtures of sw with pelletised coffee husk (ch), rice husk (rh) or water hyacinth (wh), respectively.'**

Figures 4 and 5: As mentioned previously, the issue with the steps in the emission factors needs to be resolved. It is also not clear what is the purpose of the insets at low supersaturation.

**The steps in emission factors are a results of cases where a large fraction of a soot mode activates within a small step in supersaturation. Further above, we provide explanations for such observations. The insets are included to illustrate differences at low supersaturations – supersaturations of atmospheric relevance.**

Figure 5: 'For most other PAM experiments. . .' This analysis needs to be in the main text rather than the figure caption.

**The comment has been moved to the main text in sec. 4.6**

Figure 6b: The upper part of the label for gray-scale bar has been cut.

**That was changed already in the ACPD paper.**

Table 3: Add the chemical analysis used to determine the ash content in the figure caption.

**The following has been added to the Table caption: "The results were obtained from application of the standard protocols EN 14775, EN 15289, EN 15290 and EN 15297."**

References:

Bostrom, D., Skoglund, N., Grimm, A., Boman, C., Ohman, M., Brostrom, M., and Backman, R.: Ash transformation chemistry during combustion of biomass, Energy Fuels, 26, 85–93, 2011.

Dusek et al.: Water uptake by biomass burning aerosol at sub- and supersaturated conditions: closure studies and implications for the role of organics, Atmos. Chem. Phys., 11, 9519–9532, 2011 Obaidullah, M., Bram, S., Verma, V., and De Ruyck, J.: A review on particle emissions from small scale biomass combustion, Int. J. Renew. Energy Res., 2, 147–159, 2012.

Obernberger, I., Brunner, T., and Bärnthaler, G.: Chemical properties of solid biofuels-significance and impact, Biomass Bioenerg., 30, 973–982, 2006.

Reavell, K., Hands, T., and Collings, N.: A fast response particulate spectrometer for combustion aerosols, SAE Transactions, pp. 1338–1344, 2002.

Symonds, J. P., Reavell, K. S. J., Olfert, J. S., Campbell, B. W., and Swift, S. J.: Diesel soot mass calculation in real-time with a differential mobility spectrometer, J. Aerosol Sci., 38, 52–68, 2007.

Wittbom et al. (2014) Cloud droplet activity changes of soot aerosol upon smog chamber ageing, Atmos. Chem. Phys., 14, 9831–9854, 2014.

---

## Author Comment (AC2) · 5 Mar 2021

We thank the anonymous reviewer for the comments and suggestions. We find that the revised version of the manuscript has improved due to these comments, and we thank the anonymous reviewers in the revised version of the manuscript. Our reponses appear in **bold** below.

The manuscript discusses the properties of cloud condensation nuclei (CCN) emitted from biomass burning of solid fuels (seven different fuels) in different cookstoves (four

[Figure]

different stoves). This study covers particle number size distribution, mixing state, particle density, chemical composition, CCN activation and particle hygroscopicity properties at the same time. The measurement results offer valuable insights for field measurements and global models regarding biomass burning particles. Overall, the paper is well written and relevant to ACP. I recommend publication after the following comments are satisfactorily addressed:

Major comments:

1. There were a bunch of measurements clearly explained in the manuscript. But I think it would be nice to have an overview plot or measurement setup sketch. It should include biomass burning setup, chambers, sampling line, and instruments. It will help readers to understand your measurements better. A simple example can be found in Smith et al., (2019).

**The requested schematic of the experimental setup has been included as the first figure in the revised manuscript.**

2. Did you consider the particle wall losses and particle loss inside the inlets (diffusion, deposition, etc.)? For example, the emitted aerosols were injected into a chamber for 10-40 minutes. What is the wall loss affection of the size distribution?

**When it comes to potential particle losses, then different effects dominate in different particle size ranges.**

**Supermicron particles**
**Potential particles with sizes of $\smile$1 $\mu$m and larger will not be captured in our set up, and while such particles may contribute significantly to the emitted particulate matter (PM), they are unlikely to influence the CCN population on a number**

basis.

**Potential diffusional particle losses**
We did not correct for diffusional losses of the very smallest particles in the sampling lines, and there are two main reasons why we would not expect such potential losses to significantly influence the reported CCN results.

I: The fast particle analyser sampled from the flue gas with a high flow rate (8 slpm) to minimize diffusional losses, and II: diffusional losses are more pronounced for the very smallest particles, which are not large enough neither to act as CCN nor to contribute significantly to the estimated PM. Hence, we do not expect any minor particle losses to influence any of the main results presented.

**Effects of aerosol storage**
Aerosol storage in the chamber did lead to particle losses, but we do not expect that to influence the reported results, since the CCN and the effective density results were not sensitive to the absolute particle concentration of the studied quasi-monodisperse populations, and the associated spectra were obtained rather fast (2-5 minutes). Wall losses and coagulation effects posed a challenge when it came to studying the effect of photochemical ageing – after some time of aerosol storage. Thus, we only discuss the results of photochemical ageing of the soot mode on a qualitative level in the manuscript.

For clarification, we have added the following sentence:
"It is worth noting that none of the listed potential minor effects will have any substantial influence on the main results reported in this study." To the previous L. 277.

In addition, we have changed L.522-523 from:
"The characteristics of the experimental setup related to wall losses and coagulation (dilution rates) may to some extent influence the inferred CCN emission factors on a quantitative level."

**To:**
**"The characteristics of the experimental setup related to losses in sampling lines and coagulation (dilution rates) may to some extent influence the inferred CCN emission factors on a quantitative level."**

**And a similar change has been made to L. 524-526.**

Minor comments:
Line 205: I could not get why "aerosol particles present in the flue gas and initially injected into the aerosol storage chamber as freshly formed or primary, while particulate matter formed in the flow reactor will be considered secondary aerosol.". Could you please give more explanation about your definition of freshly formed and secondary aerosol?

**Our definitions of primary and secondary aerosol particles are similar to the definitions applied in the literature (e.g. Obaidullah et al., 2012, Martin et al., 2013, Reece et al., 2017). We find that the inclusion of a schematic of the experimental set up has made our distinction between primary and secondary aerosol particles more clear in the revised manuscript.**

Lines 241-242: Why the soot mode particle was unaffected in the storage for up to 60 minutes? Could you find previous studies that also support this? At least from your measurement, the aged 200 nm particles always had different (most probably higher) kappa values than primary particles.

**The geometric mean of lognormal modes fitted to the soot mode sampled from the storage chamber did not change significantly over the course of the first 60 minutes after filling in any of the experiments. In some cases we observed a tendency of increasing kappa value of the 200 nm soot particles with time – which we associate with coagulation between ultrafine and soot mode particles occurring inside the chamber. For that reason, the reported kappa values for the**

**primary particles represent the very first measurements following the chamber filling.**

**The 'aged 200 nm particles' were actively aged via sampling through the oxidation flow reactor. Those aged particles always had higher kappa-values than the 200 nm particles sampled directly from the aerosol storage chamber in all the cases when ageing was simulated. Those observations are presented in Table 5. The effect of photochemical ageing was generally observed to dominate over potential coagulation effects for the reported ageing experiments. In order to clarify the type of ageing investigated, we have in the manuscript specified that we "simulated photochemical ageing" with the oxidation flow reactor.**

Lines 269-270: It is good to adjust the PNSD for dilution rates and normalized to the corresponding consumption of dry fuel mass. I would suggest including the error bars in Fig. 1. The error bars could be 25th and 75th percentiles (with median lines) or one stand deviation (with mean lines). This will help us to understand the fluctuation of PNSD during the experiments.

**When it comes to the particle number size distributions presented in Fig. 1, we have optimized it for easy intercomparison between different combinations of stoves and fuels. We find that essential, and it is not possible to obtain that with inclusion of percentiles for the many particle number size distributions presented.**

**Instead, we provide the requested information by addition of the following paragraph to Section 4.1:**
**"We have studied the particle number size distributions over the chamber injection time windows to provide some semi-quantitative information about the variability. The particle number concentration for the diameter size bin where maxima on average were observed for the ultrafine modes varied relatively by $\pm28\%$, $\pm17\%$, $\pm9\%$ and $\pm6\%$ for the 3S, the RS, the NDS and the FDS, respectively. The**

**reported variability ranges correspond to the range from the 25th to the 75th percentile. It was not possible in general to report similarly derived variabilities for the soot-mode-maxima due to less well-defined modes in many experiments. Instead we have looked into the variability in number concentrations for the size bin centered close to 150 nm. Those variabilities were about $\pm35\%$, $\pm50\%$ and $\pm10\%$ for the 3S, the RS and the NDS, respectively, again with the ranges corresponding to the 25th to 75th percentile range. The relative soot-mode particle number variability for the FDS-sw experiment was about 14%, while it was significantly more pronounced (20%-100%) for the other FDS experiments with very modest absolute soot mode particle number concentrations. Comparison between stoves should be carried out with precaution in this context, since the relevant time intervals, fuels and number of experiments with each stove varied. Nevertheless, we observed indications of more constant ultra-fine particle number emissions with increasing combustion temperature and improved combustion temperature-stability from the 3S to the FDS. Furthermore, it is worth noting that the soot mode variability appears to be more pronounced for the RS relative to the 3s, but additional studies are needed for a robust investigation of such potential differences. The pronounced variability in aerosol emissions in particular from the 3S and the RS illustrates the need for studying average emissions over a full combustion cycle for those stoves in order to assess the general aerosol emission features. Due to the described variability in emissions, we prioritised repeats of the 3S and the RS experiments as indicated in Table 1."**

Lines 305-308: Did you see relative higher slopes of the CCN activation spectra for the FDS, sw and NDS, sw-ch? Clear ultrafine and soot modes were observed for these two types of biomass burning particles. There is an overlap of ultrafine and soot modes in the size range around 100 nm. If we assume the ultrafine and soot mode particles have different chemical compositions, a relatively higher slope would be expected.

**We optimized the CCN measurements towards obtaining CCN spectra for quasi-**

monodisperse aerosol populations very fast - in parallel with the APM measurements. We found that such an approach was essential in this study, and this approach does not provide a solid foundation for detailed studies of the CCN spectra for the complex aerosol populations studied. We invested a lot of time and effort into quality control of the CCN spectra, as described in the manuscript. This in order to ensure, that reported kappa values represented the majority of any given quasi-monodisperse aerosol population studied. We have attempted to extract additional information from the CCN spectra as described in the text. However, we do not find our data set adequate for such detailed analysis. We have added the following to the end of section 4.2: "However, it should be noted that the CCNc experimental approach was optimised for fast scans in parallel with the APM measurements. Hence, nor the DMA transfer function neither the CCNc operation were optimised for extracting information about the mixing state from the CCN spectra."

References:

Martin, M., Tritscher, T., Juranyi, Z., Heringa, M. F., Sierau, B.,Weingartner, E., Chirico, R., Gysel, M., Prévôt, A. S., Baltensperger, U., and Lohmann, U.: Hygroscopic properties of fresh and aged wood burning particles, J. Aerosol Sci., 56, 15–29, 2013.

Obaidullah, M., Bram, S., Verma, V., and De Ruyck, J.: A review on particle emissions from small scale biomass combustion, Int. J. Renew. Energy Res., 2, 147–159, 2012.

Reece, S. M., Sinha, A., and Grieshop, A. P.: Primary and photochemically aged aerosol emissions from biomass cookstoves: chemical and physical characterization, Environ. Sci. Technol., 2017.

Smith, D. M. et al. (2019) 'Construction and Characterization of an Indoor Smog Chamber for Measuring the Optical and Physicochemical Properties of Aging Biomass Burning Aerosols', Aerosol and Air Quality Research, 19(3), pp. 467–483. doi: 10.4209/aaqr.2018.06.0243

---

## Author Response (AR2)

**Editor Decision: Publish subject to minor revisions (review by editor)** (27 Mar 2021) by Paul Zieger Comments to the Author: Dear authors.**

Thank you for your revised manuscript. All comments by the referees have been satisfactorily incorporated. I have had another read of your manuscript and have a few more additional mostly minor comments.

Thank you very much for the useful comments. We address them in a point-by-point fashion below in bold. Additions to the manuscript are indicated in bold in the revised manuscript. In addition to the changes described below, we have also corrected about 5 typos in the text.

- I would remove the acronyms within the abstract (sw, ch, rh, ND, FD, etc) since they are not really used within the abstract and shortly later defined again. The later introduced acronyms are also not consequently used afterwards.

The acronyms have been deleted in the abstract as suggested.

In a few cases the acronyms FD and ND were used instead of FDS and NDS in the text (5 and 2 times, respectively). Furthermore, slightly different 'formats' had been used in 4 subfigure labels (Fig. 2+5). That has all been corrected in the revised manuscript for consistency.

We found that the FDS and the NDS acronyms were not well-defined in the text. That has changed in the revised manuscript in L. 105.

In addition, we made the following changes:

L. 550: "for the RS and birch and two different RS and casuarina experiments" has been changed to: "for the **RS-bir** and two different **RS-cas** experiments"

L. 562-63: "the NDS and sesbania pellets included" has been changed to: "with the NDS-ses included"

- Line 15: Add a space before "with". Done

- Some of the references in the introduction are selections and not complete. I would suggest to at least add "e.g." in front of those cases were just some recent and not necessarily the original studies are cited. Please also order them by year.

**All citations of multiple studies have now been ordered chronologically.**

**e.g. has been added in the following three cases:**

- L. 24: (e.g. Crutzen and Andreae, 1990)
- L. 36: (e.g. Jetter and Kariher, 2009; MacCarty et al., 2010)
- L. 39: (e.g. Bølling et al., 2009; Lamberg et al., 2011; Reece et al., 2017)

- Line 42, 43: Add "particle diameter" before "<2.5micron" and ">2.5micron"

**Done**

- Line 47: A reference is needed for the sentence "It was suggested that ..."

**"It was also suggested" has been replaced by : "Gaudichet et al., (1995) also suggested"**

- A general Latex tip (applicable within the text but also for the equations): Add "\rm" to the parts of the variable or equations where you use text. So e.g. \$\rho\_{\rm eff}\$ or {\rm with} (in Eq 1) or g/kg\$\_{\rm dry fuel}\$. Acronyms should also not be in italics (e.g. PM\_\$0.5\$)

**The suggested changes have been carried out throughout the text.**

- Line: 157: I guess you used the CCN-100 from DMT?

We have added: "(CCNc, CCN-100, Droplet Measurement Technologies)" to L. 150, and deleted "(Droplet Measurement Technologies)" from L. 157.

Also the manufacturer of the AE33 has been included in L.198: "(AE33, Magee Scientific)"

- Line 180: "slpm" -> "lpm"

"slpm" is the correct unit in this case

- Line 190: What kind of neutralizer was used for the SMPS? Was a pre-impactor installed?

**The following has been added (L.193-194): "The SMPS was operated with an advanced aerosol neutraliser (model 3088) and an inlet impactor (0.071 cm) both from TSI."**

- Line 192: It is not clear to me if you really used the SP-data from the SP-AMS. Please clarify

The statement in L. 426-428: "That is supported by the AMS measurements, where the relative increase in PM due to the simulated photochemical ageing was due to organic species with fragmentation patterns characteristic of SOA particles." relies in part on measurements of the BC concentration in the unaged/aged aerosol. The SP-AMS data from this campaign will be presented in a lot more detail in future publications, and for clarity, we find it meaningful to present the instrument as SP-AMS in this paper for consistency between the different publications based on the SUSTAINE campaign.

- Line 210-213: Could you give a reference or archive for the EN-protocols?

**We have included references to webpages in the revised version:**

L. 214: "the EN ISO 16948 protocol (www.iso.org)."

L. 217: "the EN 14775 and the EN 15289, 15290 and 15297 protocols (https://standards.iteh.ai), respectively."

- Figure 2: Why is the unit for the particle number concentration (cm^-3) gone? Please double-check your units.

Particle number size distributions reported in cm-3 would depend on the dilution in the system. Hence, we have normalized the particle number size distributions to fuel consumption as described in the text. We obtain a  $CO_2$  concentration in the units: kg( $CO_2$ )/cm3, which was converted to kg\_dry\_fuel/cm3 as described in the text. Considering the units, we obtain: cm-3/(kg\_dry\_fuel/cm3)= (kg\_dry\_fuel)-1 for the normalized particle number size distributions.

- Concerning the second major comment by reviewer 1 and the particle losses: This one additional sentence is not sufficient. Please explicitly state in the revised manuscript that particle loss calculations were not done and add your reasoning. Concerning the SMPS data: Was this also not loss-corrected?

We have replaced the previous addition: "It is worth noting that none of the listed potential minor effects will have substantial influence on the main results reported in this study." With the following more detailed description: "We did not carry out particle loss calculations in the flue gas and sampling lines. The measurements of particle number size distributions in the flue gas with the FPA were optimised for minimal particle losses, and for the size range of relevance for CCN (Dp>~30 nm), we would not expect any pronounced losses."

The SMPS data were not corrected for losses in neither the aerosol storage chamber nor sampling lines/PAM. We do not make use of the SMPS data on a direct quantitative basis at any point. It is not straightforward to model losses of vapours/particles inside PAM. Hence, we mainly make use of SMPS data on a qualitative level for the results presented in Fig. 6, where we always normalize to the soot mode (accumulation mode range), where particle losses are at a minimum.

The limitations in terms of quantitative measurements of the 'aged' CCN population is therefore described and discussed in detail in L. 611-634.

- Concerning the data, I recommend that you store the data behind your study on an open-accessible repository (with a DOI). The current way of referencing to data via a personal contact is not optimal. Please have a look at the data guidelines of ACP: https://www.atmospheric-chemistry-and-physics.net/about/data\_policy.html

We are highly interested in other researchers and particular atmospheric modelers making use of our data. In fact, we consider the current study a major step forward in estimating CCN emmisions from biomass burning. However, there are 3 main reasons why we very much would like to keep data access as 'upon request':

- 1. It is essential to us that interpretation and application of our data is carried out according to what they actually represent (e.g. not full water boiling experiments, estimated PM0.5).
- 2. We have a wide range of other supportive data available, e.g. from the aethalometer. Therefore, we would prefer to have a dialogue with potential end-users in every case since we may be able to provide different normalisations of the CCN e.g. to PM absorption or similar depending on the application.
- 3. Most importantly: There are a number of additional SUSTAINE studies in the pipeline and a planned SUSTAINE 2.0. Gradually, we learn a lot more about these fuels and physico-chemical particle properties as we progress with additional analysis. We may over the coming months

obtain pronounced new insight in essential supporting data e.g. size resolved chemical composition of PM from impactors. Hence, we will most likely be able to provide a lot more support to atmospheric modelsers, than what would be available with 'just' the data presented in this paper.

We agree that it is not ideal to have just one single data contact person, so in the revised manuscript, we have revised to: "All presented data can be requested from the corresponding author T. B. Kristensen **or J. Pagels (joakim.pagels@design.lth.se)**"

We suppose this approach is acceptable(?), based on very recent ACP studies based on laboratory measurements making use of a similar data availability approaches (e.g. Kostenidou et al., ACP, published: 26/3-2021; Takhar et al., ACP, published: 1/4-2021)

Thanks and kind regards

Paul.